# Scalable Topology-Preserving Graph Coarsening: Concepts and Algorithms

Xiang Wu [1]   Rong-Hua Li [1]   Xunkai Li [1]   Kangfei Zhao [1]   Hongchao Qin [1]   Guoren Wang [1]

## Abstract

Graph coarsening reduces the size of a graph while preserving certain properties. Most existing methods preserve either spectral or spatial characteristics. Recent research shows that topology-preserving coarsening methods maintain GNN performance on coarsened graphs but suffer from exponential time complexity. To address these problems, we propose Scalable Topology-Preserving Graph Coarsening (STPGC) by introducing the concepts of graph strong collapse and graph edge collapse extended from algebraic topology. STPGC comprises three new algorithms, *GStrongCollapse*, *GEdgeCollapse*, and *NeighborhoodConing* based on these two concepts, which eliminate dominated nodes and edges while rigorously preserving topological features. We further prove that STPGC preserves the GNN receptive field and develop approximate algorithms to accelerate GNN training. Experiments on node classification with GNNs demonstrate the efficiency and effectiveness of STPGC.

## 1. Introduction

Graph-structured data is ubiquitous in real-world scenarios, spanning applications such as social networks, recommender systems, and molecular graphs. As data volumes surge, analyzing large-scale graphs poses significant computational challenges. One prominent class of techniques to tackle this problem focuses on reducing graph size, including graph coarsening (Kumar et al., 2023; Han et al., 2024), graph sparsification (Chen et al., 2021a; Hui et al., 2023), and graph condensation (Jin et al., 2021; Fang et al., 2024). Among these, graph coarsening has attracted considerable attention due to its solid theoretical foundations and practical effectiveness. It generates a downsized graph by merging specific nodes while preserving certain graph properties. Its most prevalent application is scaling up GNNs by

training them on the coarsened graphs while maintaining performance on the original graphs (Huang et al., 2021).

Most graph coarsening methods focus on preserving spatial or spectral features. Spatial approaches retain structural patterns like influence diffusion or local connectivity (Purohit et al., 2014), while spectral methods target the graph Laplacian's eigenvalues or eigenvectors (Loukas & Vandergheynst, 2018). These features are typically maintained by optimizing metrics such as reconstruction error (LeFevre & Terzi, 2010) or relative eigenvalue error (Loukas, 2019). Additionally, greedy pairwise contraction of nodes has been shown to preserve spectral properties (Chen & Safro, 2011).

Despite the effectiveness of these methods, none of them preserves the *topological* features of graphs. Here, the topological features stem from the theory of algebraic topology (Eilenberg & Steenrod, 2015), characterizing the invariants of data under continuous deformation, such as stretching and compressing (Dey & Wang, 2022). Specifically, graph topological features encompass connectivity, rings, and higher-order voids. These features capture the essential information of graphs and have been shown to benefit downstream applications (Yan et al., 2022a; 2023; Zhu et al., 2023; Immonen et al., 2023; Meng et al., 2024b). For example, ring information has improved the performance of graph representation learning (Horn et al., 2022; Hiraoka et al., 2024; Nuwagira et al., 2025), especially in biology and chemistry (Townsend et al., 2020; Aggarwal & Periwal, 2023). Additionally, leveraging local topological structures contributes to improved node classification (Chen et al., 2021b; Horn et al., 2022), and link prediction (Yan et al., 2021; 2022b). Furthermore, recent research has shown that preserving topological features in graph coarsening benefits node classification with GNNs, while disrupting topological structures degrades performance (Meng et al., 2024a).

While the importance of topological structures is well-established, the question of how to effectively preserve them in graph coarsening remains largely unexplored. The only existing method, Graph Elementary Collapse (GEC) (Meng et al., 2024a), relies on clique enumeration to identify reducible structures. This incurs an exponential worst-case time complexity of $O(3^{n/3})$, rendering it unscalable for large graphs. To bypass this, GEC first partitions graphs into subgraphs for separate processing, applies elementary collapse to each subgraph independently, and finally recon-

[1]Department of Computer Science, Beijing Institute of Technology. Correspondence to: Rong-Hua Li <lironghuabit@126.com>.

*Proceedings of the 43rd International Conference on Machine Learning*, Seoul, South Korea. PMLR 306, 2026. Copyright 2026 by the author(s).

structs the coarsened graph by stitching these processed components back together. However, this workaround poses two major limitations: (i) the partitioning process inevitably disrupts global topological features, which cannot be recovered during reconstruction; and (ii) the exponential complexity persists within subgraphs, while the additional reconstruction overhead hinders its utility on large-scale graphs.

To address these problems, we draw inspiration from strong collapse and edge collapse in algebraic topology (Boissonnat & Pritam, 2019; 2020; Barmak & Minian, 2012). By extending these concepts to graphs, we introduce scalable topology-preserving graph coarsening (STPGC), consisting of three graph coarsening algorithms rigorously preserving the topological features. Specifically, we first propose *GStrongCollapse* and *GEdgeCollapse*, which iteratively identify dominated nodes and edges and reduce them, respectively. To enable further coarsening ability, we propose the *neighborhood coning* algorithm to insert dominated edges between the neighbors of nodes to create new dominated nodes and further reduce them with graph strong collapse. These algorithms are more efficient than GEC because they eschew costly clique enumeration by directly identifying reducible nodes and edges through neighborhood inclusion. We demonstrate the effectiveness of STPGC for maintaining GNN performance from the perspective of preserving the GNN receptive field. To maximize scalability for GNN training, we extend the framework with *ApproximateCoarsening* that relaxes dominance conditions for flexible coarsening ratios. This strategy effectively balances topological preservation with the algorithmic efficiency.

To summarize, our contributions are: (1) **Novel concepts and algorithms**: We introduce the concepts of graph strong collapse and graph edge collapse, extended from algebraic topology to graph analysis. Building on the concepts, we propose three scalable, topology-preserving graph coarsening algorithms. (2) **Important application**: We apply STPGC to accelerate GNN training. We prove that STPGC preserves the GNN receptive field on the coarsened graph, and propose *ApproximateCoarsening* for scalable GNN training. (3) **SOTA performance**: Extensive experiments demonstrate that STPGC outperforms state-of-the-art (SOTA) approaches in node classification while delivering up to a 37x runtime improvement over GEC.

## 2. Preliminaries

### 2.1. Notations and Concepts

**Graph Coarsening.** We denote a graph as $\mathcal{G} = (\mathcal{V}, \mathcal{E})$, where $\mathcal{V}$ and $\mathcal{E}$ are the sets of $n$ nodes and $m$ edges. If the nodes have $d$-dimensional features and labels, $\mathbf{X} \in \mathbb{R}^{n \times d}$ represents the feature matrix of the nodes, and $\mathbf{Y} \in \mathbb{N}^n$ represents the labels of the nodes. Graph coarsening aims to derive a graph $\mathcal{G}^c = (\mathcal{V}^c, \mathcal{E}^c)$ and the corresponding $\mathbf{X}^c$ and $\mathbf{Y}^c$, where $|\mathcal{V}^c| << |\mathcal{V}|$, while preserving the key characteristics of $\mathcal{G}$.

**Simplex and Simplicial Complex.** A $k$-simplex $\tau$ is strictly defined as the convex hull of $k + 1$ affinely independent points (Dey & Wang, 2022). In the graph domain, this corresponds to a clique of $k + 1$ nodes: 0-, 1-, and 2-simplices represent nodes, edges, and triangles, respectively. A collection of simplices forms an abstract simplicial complex $\mathcal{K}$ if it satisfies the hereditary property: for any $\tau \in \mathcal{K}$, all subsets (faces) of $\tau$ are also included in $\mathcal{K}$ (Boissonnat & Pritam, 2020). Specifically, a complex constructed from the cliques of a graph is termed a **clique complex**.

**Elementary Collapse (Whitehead, 1939).** Given a $k$-simplex $\tau$, a simplex $\sigma \subseteq \tau$ is a face of $\tau$. A simplex is maximal if it is not contained in any other simplex. A face $\sigma$ is termed a free face if it is strictly contained in a unique maximal simplex $\tau$. Removing the pair $(\sigma, \tau)$ is called an elementary collapse. Graph Elementary Collapse (GEC) (Meng et al., 2024a) extends elementary collapse to graphs by considering a $(k + 1)-$clique as a $k-$simplex, transforming a graph to a clique complex. GEC enumerates all cliques and identifies inclusion relationships. However, since clique enumeration has a worst-case time complexity of $O(3^{n/3})$, it becomes impractical for large-scale graphs. To address this, we introduce two scalable operators called graph strong collapse and graph edge collapse.

### 2.2. Graph Strong and Edge Collapses

We then introduce the concept of graph strong collapse and graph edge collapse. We observe that the strong and edge collapses originally defined for simplicial complexes (Boissonnat & Pritam, 2019; 2020) are directly applicable to nodes and edges (i.e., low-dimensional simplices). Motivated by this, we extend these to the graph domain. We first define the relevant neighborhood concepts and then formally define graph strong collapse and edge collapse.

The *open neighborhood* of a node $u \in \mathcal{V}$ is defined as $N(u) = \{v \in \mathcal{V} | (u, v) \in \mathcal{E}\}$. We denote $deg(u) = |N(u)|$ as the degree of $u$. The *closed neighborhood* of $u$ is defined as $N[u] = \{u\} \cup N(u)$. The *open and closed neighborhood of an edge* $(x, y) \in \mathcal{E}$ is defined as $N(x, y) = N(x) \cap N(y)$ and $N[x, y] = N[x] \cap N[y]$, respectively.

**Definition 2.1.** (Graph strong collapse) On a graph $\mathcal{G}$, $u$ is said to be dominated by a node $v$ if and only if $N[u] \subseteq N[v]$, and $u$ is called a **dominated** node if it is dominated by another node. Removing a dominated node and its connected edges from $\mathcal{G}$ is called a **graph strong collapse**.

**Example 2.2.** In Figure 1(a), the red leaf nodes are dominated by their parent nodes. We can remove the red nodes with graph strong collapse and their parent nodes then be-

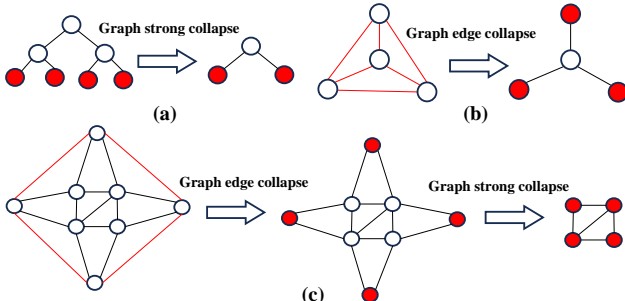

*Figure 1.* Examples of graph strong collapse and graph edge collapse. The dominated nodes and edges are shown in red.

come dominated, which can be further removed via graph strong collapse.

**Definition 2.3.** (Graph edge collapse). An edge $(x, y)$ is said to be dominated by a node $v$ if and only if $N[x, y] \subseteq N[v]$, where $v \notin \{x, y\}$. $(x, y)$ is called a **dominated** edge if it is dominated by another node. Removing a dominated edge from $\mathcal{G}$ is called a **graph edge collapse**.

**Example 2.4.** An example of dominated edges is illustrated in Figure 1(b), where all red edges are dominated. The three outer red edges are dominated by the central node, while each of the three inner red edges is dominated by one of the outer nodes. Reducing a dominated edge is a graph edge collapse. Once the three outer edges are eliminated, the inner edges are no longer dominated.

**Homotopy.** A critical property of graph strong and edge collapses is that they preserve the *homotopy equivalence* between the original and the reduced graph. Two graphs are defined as homotopy equivalent if their underlying clique complexes can be continuously deformed into one another (Dey & Wang, 2022). Crucially, preserving this equivalence ensures that the reduced graph retains the same topological features as the original, including connected components, rings, voids, and Betti numbers, thereby capturing essential local and global connectivity structures (Meng et al., 2024a). This property is formally established in the following Lemma (proofs are deferred to the Appendix A).

**Lemma 2.5.** *(Homotopy Equivalent) Let $\mathcal{G}^c$ be a subgraph derived from $\mathcal{G}$ through graph strong collapses and graph edge collapses, then $\mathcal{G}^c$ and $\mathcal{G}$ are homotopy equivalent.*

## 3. Graph Coarsening Algorithms

In this section, we present three topology-preserving graph coarsening algorithms: *GStrongCollapse*, *GEdgeCollapse*, and *NeighborhoodConing*. These algorithms are employed collaboratively for the graph coarsening task. We then introduce their application in accelerating GNN training.

---

**Algorithm 1: GStrongCollapse**

**Input:** $\mathcal{G} = (\mathcal{V}, \mathcal{E})$.
**Output:** The coarsened graph $\mathcal{G}^c = \{\mathcal{V}^c, \mathcal{E}^c\}$.
1   *NodeQueue* $\leftarrow$ *push* all the nodes in $\mathcal{V}$;
2   $\mathcal{G}^c \leftarrow \mathcal{G}$;
3   **while** *NodeQueue and the coarsening ratio is not achieved* **do**
4      $u \leftarrow pop(NodeQueue)$;
5      $N[u] \leftarrow$ the closed neighborhood of $u$;
6      **if** $deg(u) \leq \theta_1$ **then**
7         **for** $v \in N(u)$ **do**
8            **if** $deg(v) \geq deg(u)$ **then**
9               $N[v] \leftarrow$ the closed neighborhood of $v$;
10               **if** $N[u] \subseteq N[v]$ **then**
11                  *delete $u$ from $\mathcal{G}^c$*;
12                  *Push* each node in $N(u)$ into *NodeQueue* if it's not in *NodeQueue*;
13                  **break**;

14   **return** $\mathcal{G}^c$

---

### 3.1. Graph Strong Collapse

*GStrongCollapse* progressively coarsens the graph by iteratively merging dominated nodes $u$ into their dominators $v$ to form supernodes. Algorithm 1 implements this efficiently using a queue to track candidates. For each node $u$, we scan its neighbors (lines 7–10); if a dominator $v$ is identified, $u$ is merged into $v$, and the search terminates early. Crucially, as removing $u$ may alter local dominance relations, its neighbors $N(u)$ are re-queued for reevaluation (line 12).

To improve efficiency, we employ several strategies. First, we expedite checks using the necessary condition $deg(v) \geq deg(u)$ (line 8) and neighbor hash tables for fast lookup. Second, to avoid the $O(|N(u)| + |N(v)|)$ cost of sparse matrix modification, we use lazy deletion, marking nodes as removed in $O(1)$ time. Finally, we skip nodes with degrees exceeding a threshold $\theta_1$, as high-degree nodes are rarely dominated. It is straightforward that Algorithm 1 faithfully implements the process defined in Definition 2.1. The time and space complexity of Algorithm 1 are $O(m\theta_1)$ and $O(|\mathcal{V}| + |\mathcal{E}|)$, respectively. Due to space constraints, the detailed complexity analysis is deferred to Appendix B.

### 3.2. Graph Edge Collapse

While *GStrongCollapse* is effective on reducing simple structures like trees and cliques, it struggles with complex graphs lacking initially dominated nodes. An example is shown in Figure 1(c): initially, no node is dominated. However, applying *GEdgeCollapse* to remove the dominated red edges breaks the deadlock, exposing new dominated nodes for subsequent reduction. Algorithm 2 achieves this by iteratively deleting dominated edges. Upon deletion, adjacent edges are re-queued to reflect neighborhood updates. For efficiency, we skip edges $(x, y)$ for which $deg(x) + deg(y) > 2\theta_1$. This procedure faithfully implements Definition 2.3, thus preserving the homotopy equiva-

**Algorithm 2:** GEdgeCollapse

**Input:** $\mathcal{G} = (\mathcal{V}, \mathcal{E})$.
**Output:** The coarsened graph $\mathcal{G}^c = \{\mathcal{V}^c, \mathcal{E}^c\}$.
1   *EdgeQueue* $\leftarrow$ *push* all the edges in $\mathcal{E}$;
2   $\mathcal{G}^c \leftarrow \mathcal{G}$;
3   **while** *EdgeQueue* **do**
4      $(x, y) \leftarrow pop(EdgeQueue)$;
5      **if** $deg(x) + deg(y) \leq 2\theta_1$ **then**
6         $N(x, y) \leftarrow$ the open neighborhood of $x$ and $y$;
7         **for** $v \in N(x, y)$ **do**
8            **if** $N(x, y) \subseteq N[v]$ **then**
9              *delete* $(x, y)$ from $\mathcal{G}^c$;
10              *Push* each edge connected to $x$ or $y$ into *EdgeQueue*, if it's not in *EdgeQueue*;
11              **break**;

12   **return** $\mathcal{G}^c$

---

**Algorithm 3:** NeighborhoodConing

**Input:** $\mathcal{G} = (\mathcal{V}, \mathcal{E})$.
**Output:** The coarsened graph $\mathcal{G}^c = \{\mathcal{V}^c, \mathcal{E}^c\}$
1   *NodeQueue* $\leftarrow$ Initialize a priority queue of nodes with ascending degrees;
2   $\mathcal{G}^c \leftarrow \mathcal{G}$;
3   **while** *NodeQueue and the coarsening ratio is not achieved* **do**
4      $u \leftarrow NodeQueue.pop()$;
5      **if** $deg(u) \leq \theta_1$ **then**
6         **for** $v \in N(u)$ **do**
7            *Inserted* $\leftarrow$ True;
8            *InsertEdgeList* $\leftarrow$ an empty list;
9            **for** $w \in N(u) \setminus \{v\}$ **do**
10              **if** *(v,w) does not exist* **then**
11                 **if** $(v, w)$ *is dominated* **then**
12                    Add $(v, w)$ into *InsertEdgeList*;
13                 **else**
14                    *Inserted* $\leftarrow$ False;
15                    **break**;
16            **if** *Inserted is True* **then**
17              **break**;
18         **if** *Inserted is True* **then**
19            *Insert* edges in *InsertEdgeList* into $\mathcal{G}^c$ and *Delete* $u$;
20            *Update* $deg(v)$ in *NodeQueue*;

21   **return** $\mathcal{G}^c$

---

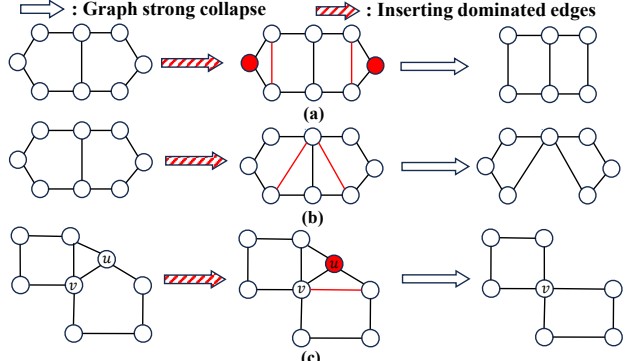

*Figure 2.* Examples of neighborhood coning. The inserted edges and newly created dominated nodes are shown in red.

lence. The time complexity of Algorithm 2 is $O(m\theta_1^2)$.

### 3.3. Neighborhood Coning

Although graph strong collapse and edge collapse effectively eliminate dominated nodes and edges, certain scenarios remain where they cannot be applied. For example, the left graph in Figure 2(a) contains neither dominated nodes nor edges, yet it can be further reduced while preserving its topological features. To explain this process, we can decompose it into two steps: first inserting dominated edges (the red edges) into the graph, and then creating new dominated nodes (the red nodes). Such a process is an *inverse* of a graph edge collapse. Here, the inverse means that after adding a dominated edge to the graph, we can obtain the original graph through a graph edge collapse, thereby guaranteeing homotopy equivalence. The red nodes can then be reduced through graph strong collapse. This approach is not restricted to creating 2-degree dominated nodes, nor is it limited to inserting just one edge (Figure 2(b), (c)). This insight naturally leads to a generalized strategy: we can create dominated nodes of arbitrary degree and subsequently reduce them, enabling further coarsening.

Building on the above analysis, we develop a transformation that induces dominance via targeted edge insertion. The central idea involves inserting dominated edges into the graph to create new dominated nodes. Designing this strategy necessitates answering two questions: where to insert these edges, and how many edges are required. First, regarding the inserting position, since dominance is predicated on neighborhood inclusion ($N[u] \subseteq N[v]$), the inserted edges must connect the candidate dominator $v$ to $u$'s neighbors. Second, regarding the number of inserted edges, ensuring full inclusion for the entire set $N(u) \setminus \{v\}$ requires inserting at most $\deg(u) - 1$ edges. This operation resembles constructing a cone with apex $v$ over the closed neighborhood of $u$, which we formalize as *Neighborhood Coning*.

**Definition 3.1.** (Neighborhood Coning) Given a node $u \in \mathcal{V}$, $u$ can be neighborhood coned by a neighbor $v \in N(u)$ if for every $w \in N(u) \setminus \{v\}$, the edge $(v, w)$ either: 1) exists in $\mathcal{G}$, or 2) can be inserted as a dominated edge. The process of adding these required dominated edges and subsequently reducing $u$ is called a neighborhood coning on $u$.

**Neighborhood Coning Algorithm.** The reduction ability of neighborhood coning is sensitive to processing order. As illustrated in Figure 2 (a) and (b), different node selections lead to varying coarsening outcomes: the upper example reduces two nodes, while the lower one reduces only a single node. Since lower-degree nodes tend to involve more complex topological features and are easier to dominate, we prioritize them by processing nodes in ascending degree order to maximize the number of reduced nodes. Algorithm 3 details this process. For each node $u$, we search for a neighbor $v$ capable of inducing dominance by inserting missing

edges to the set $N(u) \setminus \{v\}$. Upon success, $u$ is removed. In addition, any inserted edge dominated by nodes other than $u$ remains dominated after $u$'s reduction, which can also be safely reduced. Given the degree threshold $\theta_1$ and the maximum degree $d_{max}$, the worst-case time complexity of Algorithm 3 is $O(n\theta_1 d_{max}^2)$. On real-world sparse graphs, the average cost of edge dominance checks can be amortized to $O(\bar{d}^2)$ per edge, where $\bar{d}$ denotes the average node degree. Thus, the overall complexity is amortized to $O(n\theta_1^2\bar{d}^2)$, where $\bar{d} << d_{\max}$.

**Exact Coarsening.** To maximize reduction capability, we organize these three algorithms as follows. We first iteratively perform *GStrongCollapse* and *GEdgeCollapse* for $\delta_2$ iterations (lines 6-8). This loop can either continue until no further nodes or edges can be collapsed or terminate after the predefined number of iterations $\delta_2$. Following this, we apply *NeighborhoodConing* to enable further coarsening. According to Lemma 3.2, no dominated nodes remain after neighborhood coning, thereby completing the coarsening process. The procedure is named *ExactCoarsening*, as depicted in lines 5–10 of Algorithm 4.

**Lemma 3.2.** *Reducing a dominated node by neighborhood coning does not create new dominated nodes.*

### 3.4. Accelerating GNN training.

**Why STPGC preserves GNN performance.** We now apply STPGC to accelerate GNN training. To motivate this application, we first analyze why STPGC is capable of maintaining GNN performance on the coarsened graph. The core mechanism of GNNs lies in computing node embeddings by aggregating the features of a node's multi-hop neighbors, referred to as its receptive field (Ma et al., 2021). To maintain the performance of GNNs, the receptive field of each node in the original graph should be preserved as much as possible in the coarsened graph. Note that the deleted nodes are merged into supernodes. If a supernode remains within a node's receptive field, the original nodes it represents are also within that receptive field. The receptive field preservation can be characterized by the shortest path distances between nodes: if the shortest path distance between any pair of nodes does not increase after coarsening, the receptive fields of all nodes are preserved. Next, we introduce Lemma 3.3 and Lemma 3.4, which respectively show the properties of *GStrongCollapse*, *NeighborhoodConing*, and *GEdgeCollapse* in preserving the shortest path distances.

**Lemma 3.3.** *After reducing a node via GStrongCollapse or Neighborhood Coning, the shortest path distance between any pair of nodes in the remaining graph does not increase.*

**Lemma 3.4.** *After reducing an edge via GEdgeCollapse, the shortest path distance between any pair of nodes in the remaining graph increases by at most 1.*

Although *GEdgeCollapse* does not provide strict guarantees

that the shortest path distance does not increase, the proved upper bound is the tightest possible for edge deletion, since removing any edge will inevitably increase the shortest path distance between its two nodes. Therefore, the three algorithms in STPGC maximally preserve the information within each node's receptive field during the graph coarsening, thereby maintaining GNN performance. We note that the above shortest-path distance guarantees hold for the *ExactCoarsening* phase, where every reduction strictly satisfies the dominance conditions in Definitions 2.1 and 2.3.

**Approximate coarsening.** Although the above analysis shows that STPGC preserves GNN performance, the current *ExactCoarsening* cannot achieve arbitrary coarsening ratios on real-world graphs, which is necessary for graph coarsening (Huang et al., 2021). This limitation is because: (1) there exists a minimal scale for each graph below which topological features cannot be preserved, and (2) reducing a graph to this minimal core is an NP-hard problem (Eğecioğlu & Gonzalez, 1996). To address these challenges and ensure the practicability of STPGC for accelerating GNN training, we employ a two-stage strategy. We first perform *ExactCoarsening*, where topological features are strictly preserved. If the target coarsening ratio is not met, we then apply *ApproximateCoarsening*, which relaxes the collapse conditions to enable further reduction to specified coarsening ratios. The core mechanism of this approximate phase is the $r$-relaxed strong collapse. As formalized in Definition 3.5, this operator relaxes the strict inclusion requirement by allowing at most $r$ nodes in $N[u]$ to be absent from $N[v]$. Notably, the 0-relaxed case is equivalent to the exact strong collapse. Therefore, we implement the *RelaxedStrongCollapse* procedure (Algorithm 4) by adapting Algorithm 1, specifically modifying the strict dominance condition $N[u] \subseteq N[v]$ to the relaxed criteria where "$u$ is $r$-relaxed dominated by $v$."

**Definition 3.5.** ($r$-relaxed strong collapse) Given two adjacent nodes $u, v$, if $|N[v]| \geq |N[u]| > r$, and there exists a set $S_u$ with $|S_u| \leq r$ and $u \notin S_u$ such that $N[u] \setminus S_u \subseteq N[v]$, then $u$ is said to be $r$-relaxed dominated by $v$. Removing an $r$-relaxed dominated node is called an $r$-relaxed strong collapse.

When *ExactCoarsening* can no longer reduce the graph, we extend the process by replacing the graph strong collapse with the $r$-relaxed strong collapse. Specifically, we iteratively apply relaxed strong collapse and edge collapse in the same manner as *ExactCoarsening*, repeating this loop until the desired coarsening ratio is reached (lines 11-15). We do not further reapply neighborhood coning, as most chain-like structures have already been reduced, leaving little room for further reduction. As the collapse proceeds, the number of dominated nodes gradually decreases. To maintain progress, we increase $r$ by 1 whenever the number of reduced nodes in the current iteration falls below a predefined threshold $\theta_2$.

**Coarsening on Attributed Graphs.** To incorporate node

**Algorithm 4: STPGCForGNN**

---

**Input:** $\mathcal{G} = (\mathcal{V}, \mathcal{E})$.
**Output:** The coarsened graph $\mathcal{G}^c = \{\mathcal{V}^c, \mathcal{E}^c\}$.

1   $\mathcal{G}^c \leftarrow \mathcal{G}, r \leftarrow 0$;
2   $\mathcal{G}^c \leftarrow ExactCoarsening(\mathcal{G}^c)$;
3   $\mathcal{G}^c \leftarrow ApproximateCoarsening(\mathcal{G}^c)$;
4   **return** $\mathcal{G}^c$;
5   **Procedure** $ExactCoarsening(\mathcal{G}^c)$
6     **while** *the number of iterations* $< \delta_2$ **do**
7       $\mathcal{G}^c \leftarrow GStrongCollapse(\mathcal{G}^c)$;
8       $\mathcal{G}^c \leftarrow GEdgeCollapse(\mathcal{G}^c)$;
9     $\mathcal{G}^c \leftarrow NeighborhoodConing(\mathcal{G}^c)$;
10    **return** $\mathcal{G}^c$;
11   **Procedure** $ApproximateCoarsening(\mathcal{G}^c)$
12    **while** *the coarsening ratio is not reached* **do**
13      $\mathcal{G}^c \leftarrow RelaxedStrongCollapse(\mathcal{G}^c)$;
14      $\mathcal{G}^c \leftarrow GEdgeCollapse(\mathcal{G}^c)$;
15    **return** $\mathcal{G}^c$;
16   **Procedure** $RelaxedStrongCollapse(\mathcal{G}^c)$
17    Apply modified graph strong collapse by relaxing the dominance condition as $u$ is *r-relaxed dominated by* $v$;
18    **if** *number of reduced nodes* $< \theta_2$ **then**
19      $r \leftarrow r + 1$;
20    **return** $\mathcal{G}^c$

---

labels and features, we define each supernode's feature as the average of the features of all nodes mapped to it, and assign its label as the most frequent label among those nodes (Meng et al., 2024a). In labeled graphs, nodes sharing the same label are intuitively more likely to be in the same supernode. To reflect this, in Algorithm 1 and Algorithm 3, we prioritize checking neighbors in $N(u)$ that share the same label as $u$ when checking for node dominance and whether they can be neighborhood-coned by them. Furthermore, since merging similar nodes naturally reduces the graph's homophily ratio, we mitigate this discrepancy by prioritizing the removal of heterophilic edges during *GEdgeCollapse*.

**Complexity.** The total amortized time complexity of the STPGC framework is $\mathcal{T}_{total} = O\left((\delta_2 + \delta_3)m\theta_1^2 + n\theta_1^2\bar{d}^2\right)$. Here, $\theta_1$ denotes the degree threshold employed to cap the computational cost of local dominance checks for graph strong/edge collapses and neighborhood coning. The parameters $\delta_2$ and $\delta_3$ represent the maximum number of iterations performed during the *Exact* and *Approximate* coarsening phases. In practice, $\theta_1, \delta_2, \delta_3$ are treated as constants, and given that $\bar{d} \ll n$ in sparse real-world graphs, STPGC achieves linear scalability.

## 4. Experiments

**Settings.** We select five labeled benchmarks (Cora, Citeseer, DBLP, ogbn-arXiv, and ogbn-products) for node classification, and four additional unlabeled large datasets (Youtube, LiveJournal, cit-Patent, Flixster) for scalability evaluation, as summarized in Table 3. For the GNN models, we select the full-batch GNNs GCN (Kipf & Welling, 2017) and APPNP (Gasteiger et al., 2018), following prior work (Huang et al., 2021; Meng et al., 2024a), as well as the mini-

batch GNNs GraphSAGE (Hamilton et al., 2017) and Graph-SAINT (Zeng et al., 2020). We select nine graph coarsening baselines for a comprehensive comparison, including Variation Neighborhoods (Huang et al., 2021; Loukas, 2019), Variation Edges (Huang et al., 2021; Loukas, 2019), Algebraic JC (Huang et al., 2021; Loukas, 2019), Affinity GS (Huang et al., 2021; Loukas, 2019), Kron (Huang et al., 2021; Loukas, 2019), FGC (Kumar et al., 2023), UGC (Kataria et al., 2024), MPG (Joly & Keriven, 2024) and the SOTA method GEC (Meng et al., 2024a). For node classification, the GNN is trained and validated on the coarsened graph using super-node labels mapped from the original training and validation nodes, then directly applied to the original full graph for inference. All GNNs are implemented using the PyTorch Geometric framework, following the standard benchmarking protocol established by Huang et al. (Huang et al., 2021). Regarding the STPGC hyperparameters, we set the thresholds $\theta_1$ based on dataset density: 15 for Cora and Citeseer, 25 for DBLP, 50 for ogbn-arxiv, and 100 for ogbn-products. $\theta_2$ is fixed at 1% of the total node count for all datasets. The strong collapse process inherently increases graph density. To mitigate this and stabilize GNN training, we applied a DropEdge strategy on the coarsened graphs. We randomly deleted a portion of edges in the coarsened graph. Analogous to edge collapse, we prioritize the removal of heterophilic edges, setting the random drop ratio to 0.1. Code is available at https://github.com/BITNEO/STPGC.

**Node Classification.** The results are shown in Table 1, where STPGC consistently outperforms baselines across most datasets and coarsening ratios. In particular, topology-preserving methods (STPGC and GEC) outperform all other baselines, highlighting the importance of preserving topological features. Moreover, STPGC outperforms GEC by an average of 1.73% on five datasets, suggesting that STPGC more effectively preserves topological features. Although GEC is also designed for this purpose, its initial graph partition process inevitably disrupts the topological features (Figure 4). This may explain its inferior performance compared to STPGC. An exception is observed on the ogbn-arXiv dataset at $c = 0.1$ and $c = 0.2$. This may be attributed to the large number of reduced nodes, leading to loss of topological features in the coarsened graph.

**Efficiency of Different Methods.** Table 2 and Figure 3 present the runtime and memory consumption of different methods. As illustrated, STPGC consistently outperforms all baselines in both runtime and memory efficiency. In particular, compared to GEC, STPGC exhibits a much slower increase in runtime as the coarsening ratio $c$ decreases, making it more efficient for lower coarsening ratios. For instance, when $c = 0.1$, STPGC is over 15x faster than GEC on the DBLP dataset and 8.7x faster on ogbn-arXiv.

**Betti Number.** Betti numbers quantify the topological fea-

*Table 1.* Accuracy on node classification. c denotes the coarsening ratio (number of nodes in the coarsened graph/ number of nodes in the original graph). The best results are **bold**, the second best results are underlined.

| Dataset | Coarsening Method | c=1.0 | | | | c=0.5 | | | | c=0.3 | | | | c=0.1 | | | |
|---|---|---|---|---|---|---|---|---|---|---|---|---|---|---|---|---|---|
| | | GCN | APPNP | SAGE | SAINT | GCN | APPNP | SAGE | SAINT | GCN | APPNP | SAGE | SAINT | GCN | APPNP | SAGE | SAINT |
| Cora | Var. Nei. | | | | | 81.7±0.3 | 81.4±0.5 | 78.9±0.6 | 77.0±0.8 | 80.7±0.7 | 81.5±0.6 | 76.6±0.3 | 77.3±1.0 | 73.3±1.0 | 66.6±0.8 | 65.4±5.6 | 72.2±2.1 |
| | Var. Edg. | | | | | 81.0±0.4 | 82.1±0.6 | 82.0±0.8 | 79.2±0.9 | 81.0±1.0 | 80.6±0.3 | 75.7±1.0 | 48.6±4.2 | 57.3±2.3 | 79.4±0.5 | 55.0±9.8 | |
| | Alg. JC | | | | | 81.3±0.3 | 82.5±0.5 | 80.7±0.7 | 77.8±0.8 | 79.5±0.6 | 80.1±0.6 | 80.2±0.6 | 76.4±0.7 | 66.3±1.7 | 69.1±1.6 | 78.1±0.8 | 72.6±1.6 |
| | Aff. GS | | | | | 81.3±0.5 | 82.4±0.6 | 82.1±0.5 | 78.2±1.2 | 79.5±0.5 | 79.5±0.7 | 80.0±0.5 | 75.5±1.0 | 75.2±0.6 | 73.5±1.3 | 79.2±0.6 | 73.3±1.9 |
| | Kron | 80.1±0.3 | 81.7±0.3 | 79.4±0.6 | 79.7±0.7 | 81.4±0.5 | 82.6±0.8 | 80.6±0.6 | 77.4±0.7 | 79.8±0.7 | 80.1±0.7 | 80.5±0.6 | 73.8±1.1 | 65.0±1.2 | 66.7±0.9 | 77.7±0.5 | 69.0±1.4 |
| | FGC | | | | | 79.9±1.9 | 78.7±1.3 | 79.0±1.0 | 79.5±1.4 | 77.9±1.3 | 80.4±1.6 | 78.5±1.3 | 71.1±2.6 | 68.5±0.9 | 78.4±0.7 | 78.7±1.4 | |
| | UGC | | | | | 80.5±0.4 | 84.2±0.7 | 79.0±1.8 | 80.2±0.4 | 80.3±0.5 | 81.2±0.8 | 77.5±0.9 | 80.5±0.7 | 76.9±0.4 | 78.5±1.2 | 68.2±0.8 | 74.0±0.7 |
| | MPG | | | | | 80.2±1.6 | 79.6±1.7 | 79.0±0.7 | 70.0±1.9 | 74.3±1.4 | 81.0±1.2 | 73.3±0.7 | 74.9±1.4 | 73.0±0.9 | 75.0±1.2 | 71.1±0.8 | 72.1±1.5 |
| | GEC | | | | | 80.7±0.4 | 81.3±0.7 | 79.4±1.3 | 77.9±0.9 | 80.9±0.4 | 82.0±0.7 | 80.3±1.1 | 76.7±1.7 | 80.8±0.7 | 81.7±0.5 | 80.4±1.2 | 78.7±1.1 |
| | STPGC | | | | | 82.8±0.4 | 82.8±0.3 | 82.2±0.4 | 81.2±1.0 | 82.3±0.5 | 82.9±0.2 | 81.7±0.3 | 80.9±0.7 | 82.5±0.3 | 84.5±0.3 | 82.3±0.7 | 81.1±1.3 |
| Citeseer | Var. Nei. | | | | | 71.5±0.7 | 71.7±0.6 | 70.8±0.5 | 70.4±1.1 | 70.4±0.4 | 71.1±0.4 | 69.6±0.9 | 68.2±1.3 | 56.6±0.6 | 58.2±0.6 | 61.4±1.3 | 66.0±1.5 |
| | Var. Edg. | | | | | 72.2±0.5 | 71.6±0.6 | 71.1±0.6 | 71.1±1.0 | 70.1±0.6 | 71.6±0.5 | 70.1±0.7 | 69.9±0.8 | 50.6±9.7 | 50.8±9.8 | 59.3±4.4 | 60.0±3.7 |
| | Alg. JC | | | | | 71.2±0.5 | 71.4±0.8 | 70.1±1.1 | 69.3±1.6 | 70.2±0.4 | 72.4±0.5 | 69.7±0.8 | 68.8±1.8 | 60.7±5.9 | 60.9±6.6 | 63.7±2.6 | 68.1±1.2 |
| | Aff. GS | | | | | 70.4±0.7 | 71.3±0.4 | 70.5±1.0 | 69.7±1.6 | 70.3±0.5 | 71.2±0.5 | 69.2±1.0 | 69.2±1.1 | 63.7±5.6 | 62.8±6.1 | 69.0±0.9 | 67.8±1.8 |
| | Kron | 71.6±0.8 | 71.1±0.4 | 70.7±0.5 | 69.1±0.6 | 72.2±0.5 | 72.1±0.3 | 70.3±0.6 | 70.3±0.6 | 70.4±0.5 | 71.7±0.5 | 70.4±0.5 | 70.0±0.7 | 65.8±1.6 | 66.4±0.5 | 68.5±1.6 | 67.9±0.9 |
| | FGC | | | | | 70.1±1.4 | 71.4±1.9 | 70.7±0.6 | 69.2±0.5 | 69.1±1.7 | 70.5±1.3 | 69.3±0.8 | 68.9±1.0 | 67.5±1.6 | 68.1±2.3 | 68.9±1.1 | 70.4±1.4 |
| | UGC | | | | | 69.7±1.8 | 72.5±1.0 | 68.5±1.8 | 69.0±1.4 | 68.4±0.7 | 70.4±1.0 | 68.0±0.8 | 68.3±1.3 | 59.9±1.3 | 63.5±1.3 | 66.5±0.6 | 59.8±0.9 |
| | MPG | | | | | 67.5±2.4 | 64.7±1.6 | 62.9±0.9 | 69.0±2.7 | 66.6±1.6 | 68.9±1.0 | 67.6±0.7 | 66.6±1.2 | 67.2±0.7 | 67.4±1.0 | 65.1±0.6 | 67.5±0.6 |
| | GEC | | | | | 70.1±1.4 | 71.1±0.3 | 71.0±0.6 | 71.1±0.5 | 71.4±0.5 | 71.0±0.2 | 71.1±0.9 | 70.1±0.8 | 71.6±0.2 | 72.1±0.5 | 69.1±0.7 | 70.7±0.9 |
| | STPGC | | | | | 72.3±0.5 | 72.3±0.3 | 71.8±0.4 | 70.6±1.5 | 72.4±0.6 | 71.2±0.2 | 72.0±0.2 | 70.5±1.1 | 73.2±0.4 | 73.5±0.2 | 74.2±0.5 | 72.3±1.0 |
| DBLP | Var. Nei. | | | | | 82.5±0.6 | 83.4±0.5 | 80.2±0.9 | 82.3±0.7 | 81.7±0.6 | 82.4±0.6 | 78.2±0.8 | 81.3±0.5 | 79.6±0.5 | 80.4±0.5 | 71.0±3.2 | 71.5±1.4 |
| | Var. Edg. | | | | | 82.2±0.8 | 84.2±0.7 | 80.8±0.7 | 82.7±0.5 | 80.6±0.3 | 82.6±0.8 | 73.8±1.5 | 79.1±0.8 | 79.4±0.5 | 81.9±0.6 | 65.0±2.6 | 67.3±1.5 |
| | Alg. JC | | | | | 80.6±0.7 | 83.7±0.5 | 81.4±0.5 | 82.2±0.6 | 80.2±0.7 | 82.0±0.6 | 76.3±1.4 | 80.3±0.7 | 78.1±0.8 | 79.5±0.5 | 70.7±1.6 | 71.4±1.6 |
| | Aff. GS | | | | | 82.1±0.5 | 84.6±0.5 | 82.2±0.6 | 82.5±0.3 | 81.1±0.5 | 82.3±0.7 | 78.2±1.5 | 80.9±0.5 | 79.2±0.6 | 80.2±0.4 | 68.3±1.4 | 67.8±0.9 |
| | Kron | 79.9±0.2 | 81.8±0.1 | 85.1±0.4 | 80.1±0.4 | 80.6±0.6 | 84.0±0.4 | 81.6±0.5 | 82.0±0.7 | 80.6±0.7 | 81.9±0.8 | 75.6±1.5 | 80.0±0.7 | 77.8±0.4 | 78.8±0.8 | 70.4±1.2 | 68.8±1.8 |
| | FGC | | | | | 82.0±0.3 | 83.9±0.6 | 81.4±0.5 | 82.3±0.6 | 81.8±0.7 | 83.2±0.6 | 80.3±0.4 | 80.6±0.9 | 81.8±0.9 | 82.1±0.6 | 78.4±1.1 | 77.2±1.4 |
| | UGC | | | | | 84.5±0.1 | 85.6±0.2 | 82.3±0.5 | 85.2±0.1 | 83.0±0.2 | 82.3±0.2 | 80.6±0.7 | 83.5±0.1 | 74.3±0.3 | 72.1±0.6 | 64.7±0.7 | 73.3±0.5 |
| | MPG | | | | | 84.8±0.8 | 84.8±0.5 | 83.4±0.3 | 83.2±0.6 | 80.3±0.8 | 80.3±0.5 | 77.9±0.6 | 79.4±0.5 | 75.0±0.8 | 74.5±0.6 | 71.9±0.6 | 74.9±0.5 |
| | GEC | | | | | 83.0±0.7 | 84.3±0.3 | 81.7±0.7 | 83.2±0.4 | 82.4±0.6 | 83.4±0.2 | 83.2±0.6 | 83.1±0.6 | 80.5±2.9 | 78.9±0.8 | 81.5±0.6 | 82.9±0.7 |
| | STPGC | | | | | 83.9±0.2 | 85.0±0.1 | 83.0±0.4 | 82.0±0.9 | 84.5±0.1 | 85.2±0.1 | 82.7±0.2 | 82.5±0.5 | 83.1±0.1 | 83.6±0.2 | 82.4±0.3 | 84.6±0.3 |
| ArXiv | Other methods | | | | | Out of memory (Over 256GB) | | | | | | | | | | | |
| | Var. Nei. | | | | | 64.5±1.8 | 54.6±1.5 | 62.3±1.6 | 61.6±2.4 | 64.8±2.5 | 59.6±0.6 | 62.1±0.7 | 63.1±0.4 | 45.4±3.4 | 51.6±2.4 | 44.1±2.1 | 50.8±2.4 |
| | Var. Edg. | | | | | 65.4±0.9 | 59.7±1.0 | 63.4±1.0 | 64.1±1.4 | 61.9±0.6 | 55.3±0.8 | 60.8±0.7 | 62.6±0.5 | 50.3±0.7 | 54.9±0.9 | 49.6±2.8 | 51.4±1.8 |
| | Alg. JC | 70.9±0.6 | 62.2±0.2 | 66.7±2.6 | 70.0±0.7 | 65.3±0.7 | 60.9±0.6 | 63.7±1.8 | 64.6±1.8 | 61.1±0.6 | 56.4±0.6 | 60.6±2.4 | 61.0±3.1 | 48.5±1.8 | 57.3±0.5 | 50.1±0.9 | 54.3±0.7 |
| | Kron | | | | | 64.4±0.6 | 61.1±0.7 | 64.2±1.2 | 64.6±0.8 | 56.1±0.6 | 58.5±0.7 | 54.9±0.6 | 54.6±0.9 | 57.7±0.5 | 58.6±0.9 | 55.3±0.6 | 57.9±0.4 |
| | GEC | | | | | 70.1±0.5 | 61.9±0.5 | 66.7±0.8 | 66.3±0.7 | 68.3±0.5 | 60.1±0.1 | 65.3±0.4 | 63.6±0.5 | 65.1±0.3 | 59.4±0.6 | 62.4±0.8 | 63.3±1.1 |
| | STPGC | | | | | 69.6±0.3 | 62.4±0.5 | 67.8±1.6 | 66.8±0.3 | 66.5±0.5 | 60.9±0.6 | 65.6±0.6 | 64.3±0.5 | 63.2±0.8 | 58.8±1.3 | 63.2±0.3 | 64.8±0.3 |
| Products | Other methods | | | | | Out of memory (Over 256GB) | | | | | | | | | | | |
| | GEC | 74.5±0.4 | 65.4±0.5 | 72.6±0.7 | 80.1±0.4 | 72.1±0.5 | 64.3±0.3 | 72.9±0.2 | 72.2±0.6 | 71.4±0.6 | 62.2±0.3 | 71.1±0.4 | 71.3±0.5 | 70.1±0.7 | 60.1±0.4 | 65.0±1.1 | 68.2±0.4 |
| | STPGC | | | | | 75.2±0.2 | 65.6±0.4 | 74.0±0.5 | 73.7±0.7 | 74.3±0.2 | 64.7±0.4 | 72.6±1.1 | 72.4±0.4 | 70.4±0.1 | 60.4±0.4 | 68.0±0.1 | 67.6±0.5 |

*Table 2.* Running time (s) of different coarsening methods.

| Dataset | Coarsening Method | c=0.5 | c=0.3 | c=0.2 | c=0.1 |
|---|---|---|---|---|---|
| DBLP | Variation Neighborhoods | 23.329 | 33.918 | 30.863 | 34.726 |
| | Variation Edges | 14.626 | 15.945 | 19.072 | 29.070 |
| | Algebraic JC | 22.048 | 34.683 | 35.978 | 44.321 |
| | Affinity GS | 23.627 | 18.394 | 19.846 | 18.638 |
| | FGC | 956.227 | 473.301 | 202.215 | 95.336 |
| | GEC | 8.261 | 18.820 | 36.026 | 100.417 |
| | STPGC | 3.649 | 4.821 | 5.519 | 6.371 |
| ogbn-arxiv | Variation Neighborhoods | 393.1 | 475.4 | 496.1 | 535.6 |
| | Variation Edges | 546.8 | 692.1 | 767.8 | 861.2 |
| | Algebraic JC | 557.9 | 710.7 | 815.1 | 995.2 |
| | Affinity GS / FGC | Out of Memory ( Over 256GB ) | | | |
| | GEC | 129.6 | 323.4 | 883.7 | 2474.5 |
| | STPGC | 75.4 | 178.6 | 207.8 | 286.3 |

tures of a graph. To evaluate how effectively different methods preserve topology, we compute the 1-Betti numbers of the coarsened graphs under varying coarsening ratios. As shown in Figure 4, STPGC strictly maintains the number of topological features as the coarsening ratio decreases before *ExactCoarsening* exits. In contrast, other methods lose topological features at the beginning of coarsening. GEC shows a slower decline compared to non-topology-preserving methods, as it employs elementary collapse within partitioned subgraphs, which retains some local topology. However, its partitioning process inherently disrupts global topology, resulting in inferior preservation. These results highlight the superior ability of STPGC to maintain topological features.

**Parameters Experiment.** We investigate the sensitivity of STPGC to the degree threshold parameters ($\theta_1$), which govern the scope of the *ExactCoarsening* phase. Figure 5 illustrates the impact of varying $\theta_1$ on GCN node classification accuracy and runtime across different target coarsening ratios ($c \in \{0.1, 0.3, 0.5\}$). As observed in the figure, increasing $\theta_1$ generally leads to improved classification accuracy across all three datasets. This suggests that allowing a broader range of high-degree nodes to participate in strong collapses facilitates the preservation of richer topological features, enhancing downstream GNN performance. However, this topological fidelity comes at the cost of increased runtime, as larger thresholds permit more extensive dominance checks. Notably, the accuracy gains tend to plateau once $\theta_1$ exceeds the degree of most nodes in the graph. Em-

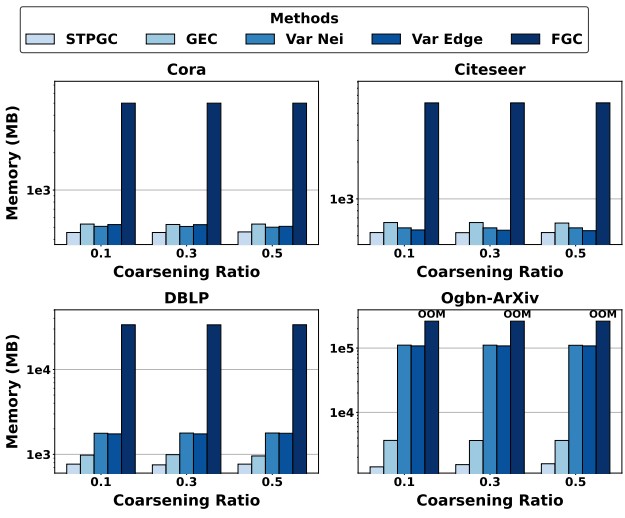

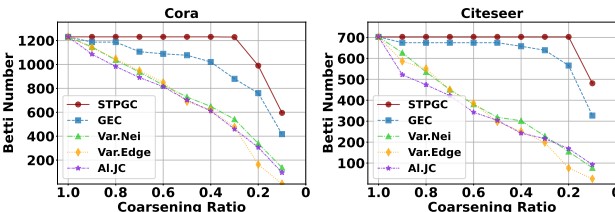

*Figure 3.* Memory overhead of STPGC and baseline methods.

*Figure 4.* Betti number preserved by different methods.

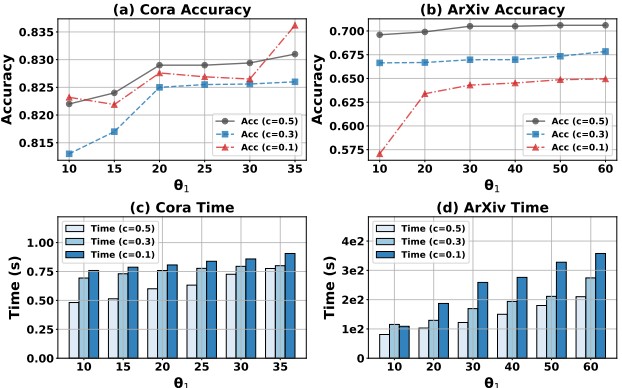

*Figure 5.* Impact of parameters on accuracy and runtime.

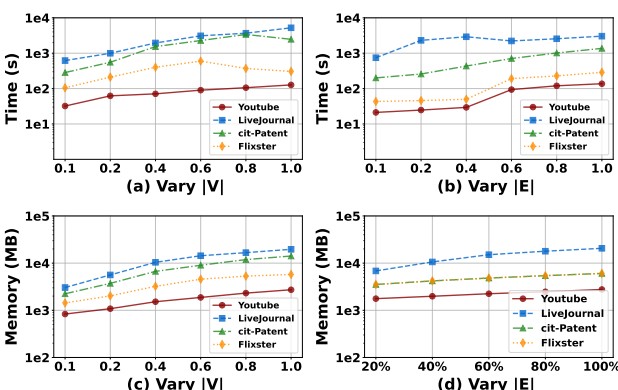

*Figure 6.* Scalability analysis of STPGC.

pirically, we find that a robust balance between accuracy and efficiency is achieved when $\theta_1$ is set to approximately 2–6 times the average degree of the respective dataset.

**Scalability Analysis.** We further evaluate the scalability of STPGCForGNN (Algorithm 4) on four large datasets (LiveJournal, Youtube, cit-Patent and Flixster). We run STPGC on subgraphs sampled at different fractions of the original graph (10% to 100% nodes), and measure runtime and peak memory on each. Subgraphs are constructed via BFS-based sampling that ensures linear growth of both $|V|$ and $|E|$ with the sampling ratio. The coarsening ratio is set to 0.3, and $\theta_1$ is fixed at 50. Figure 6 reports runtime and peak memory usage. We then compare STPGC (with varying degree thresholds $\theta_1$) with the SOTA graph coarsening method GEC (with original parameters) on the cit-Patent dataset (Figure 7). Finally, we compare STPGC with GEC on four datasets with different coarsening ratios (Figure 8). Across all datasets, memory grows monotonically with graph size, demonstrating stable space scalability. Runtime scales near-linearly for Youtube and LiveJournal throughout, and for cit-Patent and Flixster at lower sampling ratios. For the latter two, a modest runtime decline is observed at high ratios ($\geq 0.6$). This is because BFS sampling first captures the dense core, after which peripheral nodes incur negligible overhead due to degree thresholding, as confirmed by

the monotonic increase in memory usage. Compared to GEC, STPGC achieves an order-of-magnitude speedup in most configurations, and delivers up to a 37x acceleration on the cit-Patent dataset. These results validate the superior scalability of STPGC on large-scale graphs.

## 5. Related Work

**Graph Coarsening.** Existing work on graph coarsening primarily aims to preserve spectral similarity (Loukas & Vandergheynst, 2018; Bravo Hermsdorff & Gunderson, 2019). Loukas *et al.* (Loukas, 2019) introduced restricted spectral similarity to show that coarsening can approximate the eigenstructure of the original graph. More recent studies have adopted learning-based methods for data-specific coarsening. For example, Cai *et al.* (Cai et al., 2021) used deep learning to optimize edge weights through graph neural networks. Given the difficulty of training GNNs on large-scale graphs, several approaches instead focus on preserving GNN performance during coarsening. Huang *et al.* (Huang et al., 2021) demonstrated that graph coarsening not only improves GNN scalability but also acts as a regularizer to enhance training. ConvMatch (Dickens et al., 2024) minimizes the change in graph convolution between the original and

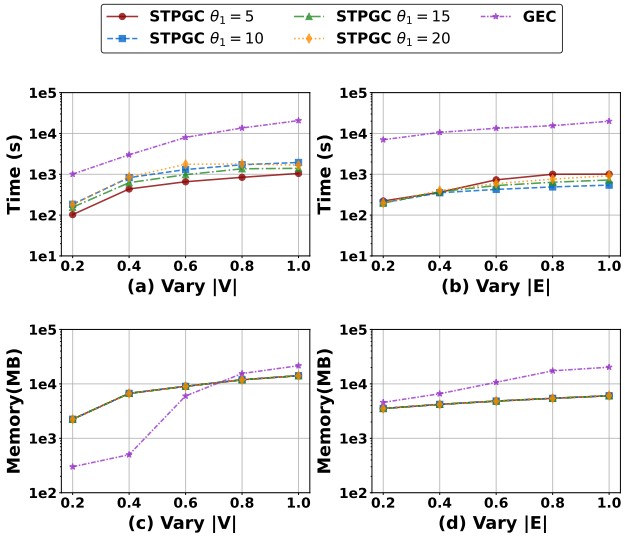

Figure 7. Scalability analysis of STPGC and GEC.

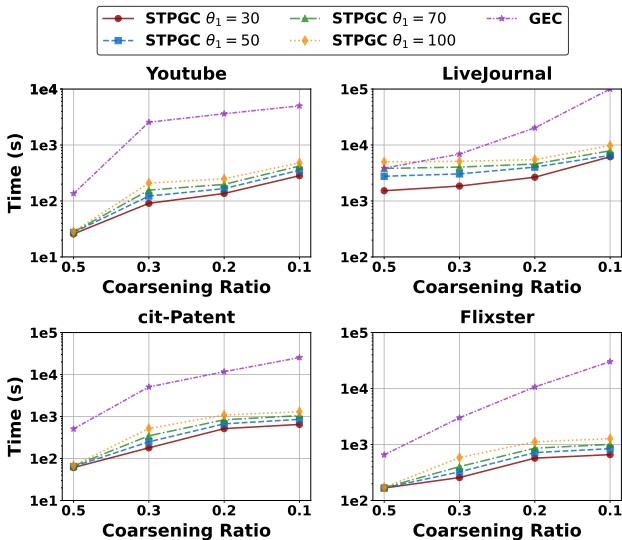

Figure 8. Results with varying coarsening ratios.

coarsened graphs. Recognizing that many real-world graphs contain node features, FGC (Kumar et al., 2023) jointly optimizes the adjacency and feature matrices to preserve both spectral and feature similarities. Finally, GEC (Meng et al., 2024a) is the only method explicitly designed to preserve topological features, employing elementary collapse to remove free faces (Whitehead, 1939).

**Dominance on Graphs.** The dominance of nodes has important applications in graph analysis. Brandes *et al.* (Brandes et al., 2017) leveraged it to compute node preorders, leading to the construction of threshold graphs (Mahadev & Peled, 1995), with applications in problems such as minimum stretch trees (Nikolopoulos & Papadopoulos, 2004) and characteristic polynomial computation (Fürer, 2017). Zhang *et al.* (Zhang et al., 2023) introduced the Neighborhood Skyline, consisting of nodes whose open neighborhoods are not included by the open neighborhood of any others. In maximum independent set computation (Chang et al., 2017; Piao et al., 2020), a node can be safely removed if its closed neighborhood fully contains another's. In shortest distance queries, neighborhood inclusion helps define equivalence classes, enabling graph compression and index size reduction (Li et al., 2019). We study new applications of dominance on graphs for accelerating GNN training.

## 6. Conclusion

In this paper, we propose a scalable and effective graph coarsening algorithm (STPGC) to preserve graph topological features. STPGC is based on graph strong collapse and graph edge collapse, ensuring a theoretical guarantee of preserved homotopy equivalence on the coarsened graph. We apply STPGC to accelerate GNN training on the coars-

ened graphs. Notably, STPGC operates as a plug-and-play preprocessing module independent of downstream GNN architectures: the graph is coarsened once, and the reduced graph can directly replace the original graph in any GNN training pipeline without modifying the model. Extensive experiments on real-world datasets demonstrate the efficiency and effectiveness of our approach. As for limitations, STPGC uses heuristic feature averaging and majority-vote label aggregation for super-nodes, which may be suboptimal for certain tasks; incorporating learnable or task-specific aggregation could further improve performance. For future directions, we plan to apply topology-preserving graph coarsening to broader machine learning domains, such as compressing biological data for drug discovery.

## Acknowledgment

This work was partially supported by the NSFC Grants U24A20255, U2241211, 62427808, 625B2021 and 62572055.

## Impact Statement

This paper presents a scalable topology-preserving graph coarsening method (STPGC) that accelerates GNN training on large-scale graphs. By producing significantly smaller coarsened graphs while preserving topological structures, STPGC reduces the computational resources and energy consumption required for downstream GNN training, contributing to "Green AI." While this work has the potential for adoption in applications such as drug discovery and biological network analysis, we do not foresee any immediate societal concerns associated with the proposed method.

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

# Appendix

**Outline.** This appendix is organized as follows:

- **Section A** provides formal proofs for all lemmas presented in the main paper.

- **Section B** presents detailed complexity analysis for each proposed algorithm: *GStrongCollapse*, *GEdgeCollapse*, and *NeighborhoodConing*.

- **Section C** describes additional experimental settings, including detailed descriptions of baseline methods, dataset statistics, and implementation details.

- **Section D** contains additional experimental results, including node classification at c=0.2, training efficiency, link prediction, graph classification, heterophilic graph experiments, and robustness to edge perturbation.

- **Section E** provides a toy graph example to compare the coarsened graphs produced by different methods.

## A. Proofs

### A.1. Proof of Lemma 2.5

*Proof.* The proof is based on the one-to-one correspondence between a 1-skeleton and its clique complex. We denote the clique complex of $\mathcal{G}$ as $\mathcal{K}$. Therefore, $\mathcal{G}$ is the 1-skeleton of $\mathcal{K}$. If $\mathcal{G}^c$ is derived from a series of graph strong collapse and graph edge collapse from $\mathcal{G}$, and $\mathcal{K}^c$ is the clique complex derived through reducing the same nodes and edges via strong collapse and edge collapse, then $\mathcal{G}^c$ is also the 1-skeleton of $\mathcal{K}^c$, as $\mathcal{K}^c$ and $\mathcal{K}$ are homotopy equivalent, we have $\mathcal{G}^c$ and $\mathcal{G}$ are homotopy equivalent. $\square$

### A.2. Proof of Lemma 3.2

*Proof.* First, after inserting an edge $(v, w)$, if $(v, w)$ still exists, it implies that the edge was solely dominated by node $u$ during the insertion. Conversely, if the edge is dominated by another node, it will remain a dominated edge after deletion, and thus be removed along with $u$.

If a new dominated node is introduced, it must be either $v$ or $w$, as only the neighborhoods $N[v]$ and $N[w]$ have changed. However, if $w$ dominates $v$ (or vice versa), the set $N(w, v)$ must be non-empty. This is because $w$ must be connected to all of $v$'s neighbors. Otherwise, both $v$ and $w$ would be 1-degree nodes, which is impossible after the graph undergoes a graph strong collapse. Let us denote the neighbors of $v$ and $w$ as $a_1, a_2, \ldots, a_n \in N[v, w]$ (Figure 9). Since $u$ dominates $(v, w)$, the nodes $a_1, a_2, \ldots, a_n$ must also be adjacent to $u$. Therefore, we conclude that $N[v] \subseteq N[u]$, which implies that $u$ dominates $v$. This leads to a contradiction, as no dominated node should remain after a strong collapse of the graph. $\square$

### A.3. Proof of Lemma 3.3

*Proof.* For node deletions performed by *GStrongCollapse* or *Neighborhood Coning*, let the deleted node be $w$. If $w$ is not in the shortest path between any two nodes $(u, v)$ in the original graph, then the shortest path distance between $u$ and $v$ remains unchanged. If a node $w$ is in the shortest path, we first prove the case in *GStrongCollapse* and then discuss that in *Neighborhood Coning*. Assume that node $w$ is dominated by node $x$. Consider any path in the original graph that contains $w$. There are two cases regarding the inclusion of $x$ in this path: **Case 1**: The path also contains $x$. Suppose the subpath in the original graph passing through $x$ and $w$ is denoted as $x, w, y$, where $y \in N[w]$. After removing $w$, since $y \in N[x]$ (implied by dominance), the path can be rerouted directly as $x, y$. This operation reduces the path length by 1. **Case 2**: The path contains $w$ but does not contain $x$. Let the subpath formed by $w$ and its neighbors be denoted as $a, w, b$. Since $N[w] \subseteq N[x]$, both neighbors $a$ and $b$ must also belong to $N[x]$. Consequently, after the removal of $w$, there exists a valid alternative path $a, x, b$. In this scenario, the length of the path remains unchanged. For the case in *Neighborhood Coning*, inserting edges does not increase the shortest path distance between nodes. After inserting edges, we still remove nodes through the *GStrongCollapse* operation; therefore, the shortest path distance does not increase.

$\square$

### A.4. Proof of Lemma 3.4

*Proof.* We denote the deleted edge as $(x, y)$ and the corresponding path as $xy$. For any pair of nodes $(u, v)$, if $xy$ is not part of their shortest path, the shortest path distance remains unchanged. However, if $(x, y)$ lies on their shortest path and is dominated by a node $w$, then the original path $xy$ can be replaced by an alternative path $xwy$. In this case, the shortest path length increases by one. Therefore, the shortest path distance between any pair of nodes $(u, v)$ increases by at most one.

$\square$

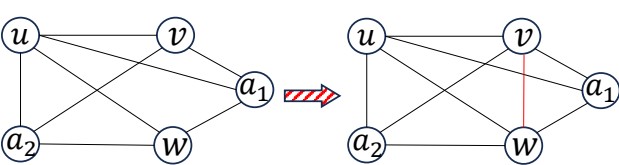

*Figure 9.* An example in the proof.

# B. Complexity Analysis

### B.1. Complexity analysis for GStrongCollapse (Algorithm 1)

In Algorithm 1, we iterate through all nodes. For a specific node $u$, we check its dominance only if $deg(u) \leq \theta_1$. The dominance check involves verifying whether $N[u] \subseteq N[v]$ for each neighbor $v \in N(u)$. Using hash sets, the intersection check for a pair $(u, v)$ takes $O(\min(deg(u), deg(v)))$ time. Since we only process $u$ where $deg(u) \leq \theta_1$, this cost is bounded by $O(\theta_1)$. Summing over all edges $(u, v) \in \mathcal{E}$, the total worst-case time complexity is:

$$\sum_{(u,v)\in\mathcal{E}} O(\min(deg(u), deg(v))) \leq \sum_{(u,v)\in\mathcal{E}} O(\theta_1) = O(m\theta_1). \tag{1}$$

### B.2. Complexity analysis for GEdgeCollapse (Algorithm 2)

In Algorithm 2, we iterate through all edges. An edge $(x, y)$ is processed only if $deg(x) + deg(y) \leq 2\theta_1$. The complexity is dominated by two steps:

1. Computing the common neighborhood $N(x, y) = N(x) \cap N(y)$, which takes $O(\theta_1)$ time.

2. Checking the dominance condition: we must verify if there exists a node $v \in N(x, y)$ such that $N(x, y) \subseteq N[v]$. In the worst case, this requires iterating through all $v \in N(x, y)$ (at most $\theta_1$ nodes) and performing an inclusion check (taking $O(\theta_1)$ time).

Thus, the cost per edge is $O(\theta_1^2)$. Summing over all $m$ edges, the total complexity is $O(m\theta_1^2)$.

### B.3. Complexity analysis for NeighborhoodConing (Algorithm 3)

In Algorithm 3, we iterate through nodes in ascending order of degree. A node $u$ is processed only if $deg(u) \leq \theta_1$. The algorithm checks if $u$ can be coned by any neighbor $v$. This requires verifying the status of edges $(v, w)$ for all $w \in N(u) \setminus \{v\}$. The nested loops iterate through neighbors $v$ and $w$, resulting in $O(deg(u)^2)$ pairs. For each pair, we check if edge $(v, w)$ exists or is dominated. On sparse real-world graphs, the cost of checking edge dominance is amortized to $O(\bar{d}^2)$, where $\bar{d}$ is the average degree. Therefore, the complexity for one node $u$ is $O(\theta_1^2 \bar{d}^2)$. Summing over all $n$ nodes, the total amortized complexity is $O(n\theta_1^2 \bar{d}^2)$.

# C. Additional Experimental Settings

**Descriptions of Baselines.** *Variation neighborhoods* and *Variation edges* (Huang et al., 2021; Loukas, 2019) contract local neighborhoods of a node and edges, respectively. They aim to optimize local variation costs based on the graph Laplacian. *Algebraic JC* (Huang et al., 2021; Loukas, 2019) , evaluates the similarity between nodes based on the Euclidean distance

*Table 3.* Datasets for node classification and scalability analysis.

| Dataset | # Nodes | # Edges | # Features | # Classes | Ave. Deg |
|---|---|---|---|---|---|
| Cora | 2,703 | 5,429 | 1,433 | 7 | 3.88 |
| Citeseer | 3,312 | 4,732 | 3,703 | 6 | 2.84 |
| DBLP | 17,716 | 52,867 | 1,639 | 4 | 5.97 |
| Ogbn-ArXiv | 169,343 | 1,166,243 | 128 | 40 | 13.77 |
| Ogbn-Products | 2,449,029 | 61,859,140 | 100 | 47 | 50.52 |
| LiveJournal | 3,997,962 | 34,681,189 | - | - | 17.35 |
| Youtube | 1,134,890 | 2,987,624 | - | - | 5.27 |
| cit-Patent | 3,774,768 | 16,518,948 | - | - | 8.76 |
| Flixster | 2,523,386 | 9,197,338 | - | - | 7.30 |

of test vectors obtained via multiple Jacobi relaxation sweeps, guiding the construction of contraction sets. *Affinity GS* (Huang et al., 2021; Loukas, 2019) follows a similar philosophy but uses a single Gauss-Seidel sweep to compute the test vectors, offering a more efficient approximation of vertex proximity. *Kron* (Huang et al., 2021; Loukas, 2019) reduction selects a subset of vertices and applies Schur complement-based Laplacian reduction to rewire the graph structure; while this approach provides strong theoretical guarantees, but it results in denser graphs and incurs high computational cost, making it less scalable. *FGC* (Kumar et al., 2023) is an optimization-based graph coarsening framework that jointly learns the coarsened graph structure and node features while approximately preserving structural similarity to the original graph. *GEC* (Meng et al., 2024a) is a topology-preserving graph coarsening method that employs elementary collapse to maintain homotopy equivalence and topological features.

**Description of datasets.** Cora and Citeseer are citation networks from the academic domain, where nodes represent papers and edges indicate citation relationships, with labels corresponding to research topics. DBLP is a co-authorship graph from computer science publications, capturing collaboration patterns among authors. ogbn-arXiv and ogbn-products are part of the Open Graph Benchmark (OGB) [1]; the former is a citation graph of arXiv papers categorized by subject areas, while the latter is an Amazon product co-purchasing network labeled by product category. The statistics of these datasets are summarized in Table 3.

**Efficient Implementation.** In *ExactCoarsening*, it is unnecessary to check all nodes and edges for dominance beyond the first iteration, because only nodes or edges whose neighborhoods have changed can potentially change to dominated. Specifically, after the initial iteration, GStrongCollapse only examines nodes whose connected edges are removed in the previous edge collapse. Similarly, GEdgeCollapse only checks edges adjacent to nodes whose neighbors were removed during the previous GStrongCollapse.

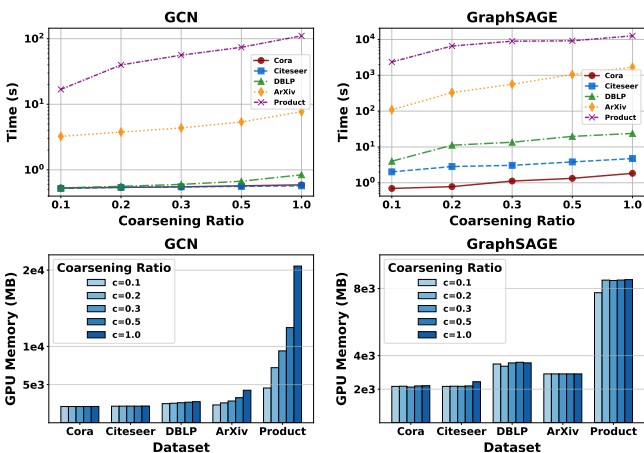

*Figure 10.* Training time and GPU memory usage of GNNs.

**Additional experimental details.** All GNNs are implemented using the PyTorch Geometric framework, following the standard benchmarking protocol established by Huang et al. (Huang et al., 2021). Regarding the STPGC hyperparameters, we set the thresholds $\theta_1$ based on dataset density: 15 for Cora and Citeseer, 25 for DBLP, 50 for ogbn-arxiv, and 100 for

---

[1] https://ogb.stanford.edu/

*Table 4.* Accuracy on node classification (c=0.2). The best results are **bold**, the second best results are underlined.

| Dataset | Coarsening Method | c=0.2 | | | |
|---|---|---|---|---|---|
| | | GCN | APPNP | SAGE | SAINT |
| Cora | Var. Nei. | 78.9±0.6 | 80.6±0.8 | 75.7±1.7 | 74.9±0.7 |
| | Var. Edg. | 73.2±1.2 | 73.0±1.5 | 80.2±0.5 | 73.1±2.2 |
| | Alg. JC | 79.4±0.7 | 81.7±0.5 | 80.0±0.5 | 76.2±1.7 |
| | Aff. GS | 80.3±0.6 | 80.5±0.7 | 79.5±0.5 | 75.0±1.1 |
| | Kron | 80.2±0.8 | 77.1±0.7 | 79.5±0.6 | 76.5±0.4 |
| | FGC | 77.5±1.9 | 76.9±2.3 | 79.7±1.6 | 77.2±1.3 |
| | GEC | 81.0±0.3 | 81.7±0.9 | 79.7±1.5 | 78.9±1.9 |
| | STPGC | **82.5**±0.4 | **84.9**±0.3 | **82.7**±0.3 | **81.7**±1.3 |
| Citeseer | Var. Nei. | 70.0±0.7 | 70.9±0.8 | 67.2±0.9 | 67.3±1.2 |
| | Var. Edg. | 54.9±0.7 | 60.6±0.7 | 69.7±1.1 | 70.2±1.5 |
| | Alg. JC | 57.1±1.6 | 67.2±1.2 | 69.7±1.0 | 68.7±0.9 |
| | Aff. GS | 69.6±0.7 | 70.7±0.7 | 69.8±1.1 | 69.1±1.2 |
| | Kron | 70.1±1.3 | 70.9±0.2 | 69.8±1.0 | 69.2±0.7 |
| | FGC | 68.1±2.3 | 69.9±2.1 | 69.4±1.0 | 68.4±0.6 |
| | GEC | 71.3±0.5 | 71.6±0.5 | 69.9±1.1 | 70.0±1.4 |
| | STPGC | **71.4**±0.6 | **71.6**±0.2 | **72.1**±0.4 | **72.1**±0.9 |
| DBLP | Var. Nei. | 80.2±0.4 | 82.3±0.8 | 75.2±1.4 | 78.2±1.2 |
| | Var. Edg. | 80.2±0.5 | 82.1±0.8 | 67.5±2.1 | 67.3±1.7 |
| | Alg. JC | 80.0±0.4 | 80.0±0.5 | 72.2±1.6 | 77.5±1.7 |
| | Aff. GS | 80.0±0.5 | 82.1±0.7 | 74.4±1.0 | 75.4±1.0 |
| | Kron | 79.6±0.4 | 80.6±0.6 | 70.7±2.6 | 76.1±1.8 |
| | FGC | 82.6±0.4 | 83.5±0.1 | 81.7±1.2 | 80.0±0.6 |
| | GEC | 83.5±1.0 | 83.8±0.2 | 82.1±0.2 | 82.5±0.6 |
| | STPGC | **84.9**±0.1 | **85.3**±0.1 | **82.4**±0.5 | **82.8**±0.5 |
| ArXiv | Var. Nei. | 53.5±1.4 | 55.3±1.0 | 52.4±2.7 | 56.7±1.4 |
| | Var. Edg. | 54.4±0.8 | 55.9±0.6 | 54.1±1.6 | 55.3±2.0 |
| | Alg. JC | 54.0±1.5 | 51.8±0.8 | 54.7±1.6 | 56.6±1.9 |
| | Aff. GS/FGC | OOM | | | |
| | Kron | 56.4±2.2 | 55.1±0.9 | 55.3±0.4 | 54.9±0.5 |
| | GEC | **67.6**±0.5 | 60.1±0.7 | 62.1±0.6 | 62.6±0.2 |
| | STPGC | 64.1±0.6 | **60.8**±1.0 | **64.9**±0.2 | **64.4**±0.3 |
| Products | Other methods | OOM | | | |
| | GEC | 70.7±0.4 | 61.1±0.4 | 69.6±0.2 | 69.1±0.3 |
| | STPGC | **73.2**±0.1 | **62.4**±0.4 | **71.4**±0.6 | 70.9±0.3 |

ogbn-products. $\theta_2$ is fixed at 1% of the total node count for all datasets. The strong collapse process inherently increases graph density. To mitigate this and stabilize GNN training, we applied a DropEdge strategy on the coarsened graphs. We randomly deleted a portion of edges in the coarsened graph. Analogous to edge collapse, we prioritize the removal of heterophilic edges, setting the random drop ratio to 0.1.

# D. Additional Experimental Results

## D.1. Node Classification at $c = 0.2$

**Additional results on other coarsening ratios.** The node classification results with a coarsening ratio of $c = 0.2$ are presented in Table 4. As shown in the table, STPGC achieves strong or competitive performance across most settings, further demonstrating its robustness and effectiveness at different coarsening levels.

## D.2. Training Efficiency

**Efficiency improvement on GNNs.** Figure 10 presents the training time and GPU memory usage of GCN (full-batch) and GraphSAGE (mini-batch) with different coarsening ratios. As shown, both the runtime and memory consumption of the two GNNs are significantly reduced on the coarsened graphs. Mini-batch training is orthogonal to graph coarsening as an approach to scaling up GNNs. Compared with GCN, the memory cost of GraphSAGE does not increase substantially with graph size, but its training time is considerably longer. In contrast, graph coarsening is a more generic approach by directly reducing the graph size and can further accelerate mini-batch GNNs when integrated with them.

### D.3. Link Prediction

We evaluate STPGC on the link prediction task using GCN with Cora, Citeseer, and DBLP at coarsening ratios $c \in \{0.5, 0.3, 0.1\}$. Table 5 reports AUC and AP (format: AUC/AP). STPGC achieves competitive link prediction performance across datasets and coarsening ratios, especially at moderate coarsening levels, and at $c = 0.5$ even surpasses the original graph on Cora and Citeseer, demonstrating that topology-preserving coarsening can benefit link prediction.

*Table 5.* Link prediction results (AUC/AP) of GCN on coarsened graphs.

| Dataset | STPGC | GEC | Var. Nei. | Var. Edg. | Alg. JC | Aff. GS |
|---------|-------|-----|-----------|-----------|---------|---------|
| *Cora (Original: 88.41/88.01)* | | | | | | |
| $c = 0.5$ | **89.62**/**89.88** | 86.11/87.20 | 80.37/81.64 | 81.61/81.75 | 79.56/79.97 | 79.31/79.92 |
| $c = 0.3$ | **88.36**/**86.82** | 86.86/84.82 | 77.90/79.93 | 77.56/77.80 | 79.75/81.53 | 77.53/79.27 |
| $c = 0.1$ | 73.76/74.48 | 68.06/71.55 | **76.31**/**76.47** | 70.40/69.29 | 75.50/76.38 | 74.35/76.52 |
| *Citeseer (Original: 90.39/91.88)* | | | | | | |
| $c = 0.5$ | **90.87**/90.36 | 88.27/88.23 | 88.96/88.94 | 90.42/**91.24** | 90.38/90.55 | 88.41/89.37 |
| $c = 0.3$ | **91.43**/91.00 | 88.86/89.65 | 88.13/87.80 | 88.28/88.89 | 91.21/90.61 | 90.61/**91.08** |
| $c = 0.1$ | 81.31/81.55 | 80.38/80.07 | **82.98**/**85.11** | 76.81/78.42 | 84.54/86.08 | 82.49/84.74 |
| *DBLP (Original: 92.33/92.52)* | | | | | | |
| $c = 0.5$ | **92.03**/**92.31** | 90.00/89.70 | 85.31/86.38 | 83.60/83.01 | 85.41/85.76 | 91.85/91.73 |
| $c = 0.3$ | 89.00/89.03 | 79.19/81.81 | 89.89/89.90 | 80.72/81.40 | 82.51/82.89 | **92.08**/**92.13** |
| $c = 0.1$ | **86.80**/**85.89** | 77.63/79.19 | 80.84/82.25 | 58.01/59.32 | 67.94/67.67 | 77.49/79.54 |

### D.4. Graph Classification

We further evaluate STPGC on graph classification using GIN (Xu et al., 2019) on MUTAG (Kriege & Mutzel, 2012) and GCN on Peptides-func (Dwivedi et al., 2022) at $c = 0.5$. As shown in Table 6, STPGC performs competitively among coarsening methods on both datasets, achieving the best result on Peptides-func and the second best on MUTAG, closely approaching the original graph performance.

*Table 6.* Graph classification results ($c = 0.5$).

| Dataset | Original | STPGC | GEC | Kron | Alg. JC | Aff. GS | Var. Nei. | Var. Edg. |
|---------|----------|-------|-----|------|---------|---------|-----------|-----------|
| MUTAG (Acc.) | 88.2 | 85.1 | 84.0 | 84.1 | 81.9 | 81.5 | 76.5 | 86.1 |
| Peptides-func (AP) | 0.183 | 0.184 | 0.173 | 0.178 | 0.175 | 0.176 | 0.178 | 0.177 |

### D.5. Experiments on Heterophilic Graphs

To evaluate STPGC on heterophilic graphs where neighboring nodes tend to have different labels, we conduct experiments on Texas, Wisconsin, and Cornell using GPRGNN (Chien et al., 2021) at $c = 0.5$. Table 7 shows that STPGC performs competitively with GEC and both topology-preserving methods generally outperform spatial and spectral baselines on Wisconsin and Cornell, suggesting that preserving global topological structures benefits heterophilic settings as well.

*Table 7.* Node classification accuracy on heterophilic graphs with GPRGNN ($c = 0.5$).

| Method | Texas | Wisconsin | Cornell |
|--------|-------|-----------|---------|
| Original | 56.5 | 52.5 | 44.1 |
| STPGC | 57.8 | **54.6** | 45.2 |
| GEC | 58.1 | 54.1 | **46.2** |
| Var. Nei. | 57.6 | 53.5 | 44.3 |
| Var. Edg. | 58.6 | 52.5 | 41.9 |
| Alg. JC | 57.0 | 53.3 | 43.2 |
| Aff. GS | 55.1 | 52.4 | 45.1 |
| Kron | **59.5** | 52.0 | 42.7 |

### D.6. Robustness to Edge Perturbation

We investigate the robustness of STPGC under random edge perturbations on Cora with GCN. Specifically, we randomly add or delete 10% and 20% of edges from the original graph before applying STPGC at different coarsening ratios. Table 8 shows that edge deletion has negligible impact on accuracy, while edge addition causes a slight performance drop.

*Table 8.* STPGC accuracy under edge perturbation on Cora (GCN).

| $c$ | Add 10% | Add 20% | Del 10% | Del 20% |
|-----|---------|---------|---------|---------|
| 0.5 | 80.1 | 78.0 | 81.3 | 81.7 |
| 0.3 | 79.7 | 78.9 | 82.6 | 82.5 |
| 0.1 | 82.9 | 82.0 | 83.0 | 83.0 |

## E. Toy Example

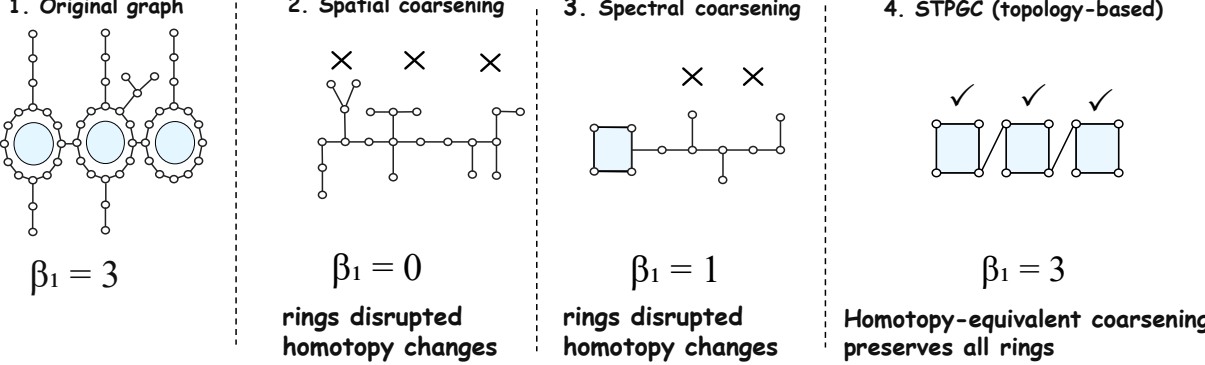

*Figure 11.* A toy example illustrating the importance of topology preservation in graph coarsening.

Figure 11 provides an intuitive illustration of why topology preservation is crucial in graph coarsening. The original graph contains three ring structures connected by short bridges, together with several contractible branches. Although spatial and spectral coarsening methods may preserve local connectivity patterns or low-frequency spectral similarity, they do not explicitly constrain the homotopy type of the graph. As a result, the three rings can be inadvertently collapsed into paths or tree-like structures, leading to the loss of 1-Betti number ($\beta_1$). In contrast, STPGC removes only topologically reducible structures, such as dominated nodes and dominated edges, while preserving irreducible global invariants. Consequently, the contractible branches are eliminated, but all three rings remain in the coarsened graph. This toy example highlights the key distinction between topology-agnostic coarsening and topology-preserving coarsening: the former may produce a smaller graph with altered global structure, whereas STPGC yields a homotopy-equivalent graph that retains the essential ring information.

