# OpenReview forum: "Scalable Topology-Preserving Graph Coarsening: Concepts and Algorithms"
_ICML.cc/2026/Conference — ICML 2026 regular_

### Official Review · Reviewer_w3GC · 2026-03-08

**Soundness:** 3
**Presentation:** 3
**Significance:** 3
**Originality:** 3
**Overall Recommendation:** 4
**Confidence:** 3

**Summary:**

This paper studies topology-preserving graph coarsening with applications to GNN training. More specifically, the authors presented three coarsening algorithms based on ideas from algebraic topology. They then apply this to accelerate GNN training with theoretical guarantee on node receptive fields, and present an approximate algorithm for practical purposes. Empirical results demonstrate the effectiveness of the proposed framework in terms of both performance and topological preservation.

**Compliance With Llm Reviewing Policy:**

Affirmed.

**Final Justification:**

I would like to thank the authors again for their engagement during the rebuttal. As a result, I maintain my positive evaluation of and original score for the paper.

**Key Questions For Authors:**

Q1. I wonder whether coarsening may assist or hinder tasks with long-range dependencies? Evaluation on the following benchmarks might provide interesting insights:

- https://arxiv.org/abs/2206.08164
- https://openreview.net/forum?id=2jf5x5XoYk
- https://arxiv.org/abs/2503.09008

Q2. How does the method perform against perturbation to input graph data, such as adversarial edge additions and deletions?

Minor: Can the authors add an explanation of how graph coarsening helps with node classification tasks, e.g. how does training and inference work in this case? For readers less familiar with this aspect of the literature this may help with the understanding of the method.

**Limitations:**

No; discussion of practical utility of the proposed framework and its limitations handling various tasks would be appreciated.

**Strengths And Weaknesses:**

Strengths:

S1. The paper investigates an important problem in graph theory and network science; the core methodology based on ideas from algebraic topology is intuitive and sound.

S2. The application of the coarsening algorithms to graph ML, theoretical analysis of receptive fields, and approximation algorithm all contribute to the practical utility of the proposed framework.

S3. Empirical evaluation is thorough in terms of both performance and behavioural analysis of the proposed method. The evaluation on non-ML tasks is especially welcome.

S4. The paper is technically sound and generally well-written.

Weaknesses:

W1. The motivation of preserving topology in GNNs can be strengthened. In addition to citing relevant literature, it would be stronger if the authors can clearly articulate (ideally via a toy example) its benefit to the task at hand.

W2. Being able to preserve the receptive fields is desirable but this alone may not guarantee good performance of GNNs. The theoretical guarantee therefore appears a little narrow. It would be good to elaborate on this point.

W3. Regarding the GNN applications, the proposed method is only evaluated on node classification tasks. It remains unclear whether it may help with a variety of tasks at link or graph level. If yes, such evaluation can further strengthen the practical utility of the method.

---

> ### Author Rebuttal · Authors · 2026-03-31
>
> **Response to Weaknesses**:
>
> **W1**: Thank you for the suggestion. We will add a visual toy example in the revision comparing coarsening methods on a graph with multiple rings. The figure will show how Spatial and Spectral methods inadvertently collapse rings into flat lines or trees, whereas our Topology-based STPGC (guided by homotopy equivalence)  preserves these irreducible global invariants.
>
> **W2**: We agree that preserving the receptive field does not solely guarantee GNN performance, which also depends on node features and labels. However, our paper focuses specifically on the topological structure. Because message-passing GNNs rely on multi-hop aggregation, proving that STPGC preserves these receptive fields structurally demonstrates its direct contribution to maintaining GNN performance. Since features and labels are not our primary focus, we adopted simple baseline processing methods. We will clarify this structural focus in the revision and discuss advanced feature/label processing as a limitation and a direction for future work.
>
> **W3**:
> The table below shows the link prediction performance (AUC / AP) of STPGC using GCN and graph classification using GIN on MUTAG. Due to the time constraints in rebuttal period, we did not finish  the experiments on complete datasets and other models. We will include additional results (e.g. PROTEINS/REDDIT-B/REDDIT-M) in revision.
> ### Cora (Orig: 88.41/88.01)
> |Ratio|STPGC|GEC|Var.Nei.|Var.Edg.|Alg.JC|Aff.GS|
> |---|---|---|---|---|---|---|
> |0.5|89.62/89.88|86.11/87.20|80.37/81.64|81.61/81.75|79.56/79.97|79.31/79.92|
> |0.3|88.36/86.82|86.86/84.82|77.90/79.93|77.56/77.80|79.75/81.53|77.53/79.27|
> |0.1|73.76/74.48|68.06/71.55|76.31/76.47|70.40/69.29|75.50/76.38|74.35/76.52|
>
> ### Citeseer (Orig: 90.39/91.88)
> |Ratio|STPGC|GEC|Var.Nei.|Var.Edg.|Alg.JC|Aff.GS|
> |---|---|---|---|---|---|---|
> |0.5|90.87/90.36|88.27/88.23|88.96/88.94|90.42/91.24|90.38/90.55|88.41/89.37|
> |0.3|91.43/91.00|88.86/89.65|88.13/87.80|88.28/88.89|91.21/90.61|90.61/91.08|
> |0.1|81.31/81.55|80.38/80.07|82.98/85.11|76.81/78.42|84.54/86.08|82.49/84.74|
>
> ### DBLP (Orig: 92.33/92.52)
> |Ratio|STPGC|GEC|Var.Nei.|Var.Edg.|Alg.JC|Aff.GS|
> |---|---|---|---|---|---|---|
> |0.5|92.03/92.31|90.00/89.70|85.31/86.38|83.60/83.01|85.41/85.76|91.85/91.73|
> |0.3|89.00/89.03|79.19/81.81|89.89/89.90|80.72/81.40|82.51/82.89|92.08/92.13|
> |0.1|86.80/85.89|77.63/79.19|80.84/82.25|58.01/59.32|67.94/67.67|77.49/79.54|
>
> ### Graph Classification on MUTAG (Ratio: 0.5)
> | Method | Original | STPGC | kron | GEC | Alg.JC | Aff.GS | Var.Nei. |
> |---|---|---|---|---|---|---|---|
> | Accuracy | 88.2 | 85.1 | 84.1 | 84.0 | 81.9 | 81.5 | 76.5 |
>
> **Response to Questions**:
>
> **Q1**:
> We evaluated graph classification on the Peptide-func dataset using a 2-layer GCN at a coarsening ratio of 0.5. Results show that STPGC still achieves the best performance among baselines. Due to limited rebuttal time, we will include these complete results including other datasets and analyses in the revision.
>
> | Method | Original | STPGC | Var.Nei. | kron | Var.Edge. | Aff.GS | Alg.JC | GEC |
> |---|---|---|---|---|---|---|---|---|
> | Test_AP | 0.183 | 0.184 | 0.178 | 0.178 | 0.177 | 0.176 | 0.175 | 0.173 |
>
>
>
> **Q2**: We conducted additional experiments on the Cora dataset using GCN. We applied random adversarial edge additions and deletions at ratios of 10% and 20%. The observation is that adversarial edge addition slightly degrades performance while edge deletion does not affect the performance. We will provide complete results and analysis in  revision.
> | Coarsening Ratio ($r$) | Add 10% | Add 20% | Del 10% | Del 20% |
> | :--- | :--- | :--- | :--- | :--- |
> | **0.5** | 80.1 | 78.0 | 81.3 | 81.7 |
> | **0.3** | 79.7 | 78.9 | 82.6 | 82.5 |
> | **0.1** | 82.9 | 82.0 | 83.0 | 83.0 |
>
> **Q3 (Minor)**: Thank you for the suggestion. We will add this explanation in the revision.
>
> **Response to Limitations**:
>   Thank you for the suggestion. We will discuss practical utility and limitations in the revision. Practical utility: STPGC  serves as a highly efficient plug-and-play graph coarsening module. Its primary utility lies in  accelerating the training  of GNNs on massive datasets while  preserving the performance.   Limitation: STPGC focuses exclusively on topological structure.  Downstream tasks that rely heavily on complex node features or labels may require additional design of feature/label aggregation strategy for optimized performance.

---

> > ### Author Rebuttal · Reviewer_w3GC · 2026-04-03
> >
> > I thank the authors for their response and most of my comments have been addressed or discussed adequately. It would still be helpful if the authors can provide more thorough experimental results as suggested. Overall I remain positive about the work and would like to maintain my positive score.

---

> > > ### Author Response · Authors · 2026-04-03
> > >
> > > Thank you for acknowledging our rebuttal. We are glad to know that our responses have addressed your concerns. As suggested, we will include the thorough experimental results in the revision. We deeply appreciate your constructive feedback!

---

### Official Review · Reviewer_4T4m · 2026-03-10

**Soundness:** 3
**Presentation:** 2
**Significance:** 3
**Originality:** 2
**Overall Recommendation:** 4
**Confidence:** 4

**Summary:**

Most existing methods preserve either spectral or spatial characteristics. To address these problems, the authors propose Scalable Topology-Preserving Graph Coarsening (STPGC) by introducing the concepts of graph strong collapse and graph edge collapse extended from algebraic topology. STPGC comprises three new algorithms, GStrong Collapse, GEdgeCollapse, and NeighborhoodConing based on these two concepts, which eliminate dominated nodes and edges while rigorously preserving topological features.

The contributions include introducing the novel concepts of graph strong collapse and graph edge collapse from algebraic topology to graph analysis. The authors propose three scalable, topology-preserving graph coarsening algorithms. They apply STPGC to accelerate GNN training and further prove that STPGC preserves the GNN receptive field on the coarsened graph. They also develop approximate algorithms to accelerate GNN training and relax dominance conditions for flexible coarsening ratios. Extensive experiments on node classification with GNNs demonstrate that STPGC outperforms state-of-the-art approaches while delivering up to a 37x runtime improvement over GEC.

**Compliance With Llm Reviewing Policy:**

Affirmed.

**Final Justification:**

I appreciate the author's response, in which they addressed numerous points regarding both the experiments and the motivation behind the work. The author supplemented their submission with corresponding graph coarsening experiments, thereby demonstrating the performance of their algorithm. Furthermore, they provided clarifications regarding the motivation, highlighting the significance of topology for GNN algorithms as well as the broader impact of this research. Although I still harbor some reservations regarding the distinction drawn between the topological and spatial levels, I consider the response as a whole to be reasonable.

**Key Questions For Authors:**

1. In scalability analysis, the author analyzed the relationship between the running time and the node or edge. In theoretical analysis, the author proved that they are linear with the time, but in Figure 6, they are represented by the exponential of (showing a nearly linear relationship with the exponential), which I think is inappropriate. Authors need to specify or give new scalability analysis diagrams.
2. It is not enough to describe the advantages of topology structure in graph coarsening. What are the advantages of preserving graph topology structure in graph coarsening compared to other graph coarsening methods, which need to carry high complexity (topology-based methods are not very common in graph coarsening), I think the authors need to further explain the decisive advantages of topology-based coarsening methods over other coarsening methods, otherwise it is hard to justify the work.
3. In the comparison algorithms, the authors compare the coarsening methods which are also based on topology and spectral methods. I think the authors need to provide more comparison algorithms, such as UGC[1] algorithm which is also based on spectrum, MPG[2] based on feature coarsening, etc. They both achieve the performance of graph coarsening while ensuring the task performance of downstream GNN.
4. I think what the author said at the motivation level is somewhat inappropriate. The author divides the existing graph coarsening methods into spatial and spectral methods, and then leads to the topological perspective provided in this article. However, I think the division between the topological perspective and the spatial perspective is conceptually ambiguous. I think the author should be able to provide more convincing evidence in this area to show that the topological perspective used in this article is a new idea.
5. In experiment settings, dropEdge strategy is adopted in the coarsened graphs. Since DropEdge is a strong regularizer that independently boosts performance, was this strategy applied equally to all baselines? To ensure a fair comparison, I strongly recommend adding an ablation study showing STPGC's results with and without DropEdge.

[1] Kataria, Mohit, and Sandeep Kumar. "Ugc: Universal graph coarsening." Advances in Neural Information Processing Systems 37 (2024): 63057-63081.

[2] Joly A, Keriven N. Graph coarsening with message-passing guarantees[J]. Advances in Neural Information Processing Systems, 2024, 37: 114902-114927.

**Limitations:**

Yes

**Strengths And Weaknesses:**

Strengths：
The paper effectively addresses the exponential time complexity of clique enumeration by creatively integrating algebraic topology concepts, specifically strong and edge collapses, into the graph coarsening pipeline. This cross-disciplinary approach significantly enhances the scalability of topology-preserving reduction, making it a viable solution for accelerating GNN training on large-scale datasets. Additionally, the manuscript is well-written, providing clear and rigorous definitions for the newly introduced topological frameworks.

Weaknesses：
The work's originality is somewhat limited as it primarily focuses on the application of existing topological tools rather than introducing fundamental new insights into graph coarsening theory. Methodologically, the reliance on relaxed approximate coarsening undermines the strict theoretical guarantees claimed. Furthermore, the technical soundness is weakened by the omission of tuning time in efficiency benchmarks and the presence of confounding factors, such as an unablated 10% DropEdge strategy, which obscures the true performance gains. Finally, the presentation is hindered by relegating essential experimental details to the appendix and providing insufficient justification for neighborhood tree aggregation through mere shortest-path metrics.

---

> ### Author Rebuttal · Authors · 2026-03-31
>
> **Response to Weaknesses**:
>
> **Originality**:
> We clarify our originality as follows: (1) **Topology Perspective.** We provide a new insight: performing graph coarsening while maintaining homotopy equivalence can preserve GNN performance on the coarsened graph, we explain why STPGC can maintain GNN performance through the preservation of the GNN receptive field. (2) **New Concepts.** We formulated graph strong/edge collapse specifically for graph analysis, adapting algebraic topology concepts. (3) **Algorithmic Contributions.**
> We proposed Neighborhood Coning (Alg. 3) to achieve enhanced reduction ability, optimized graph strong collapse and edge collapse algorithms for large graphs, and an Approximate Coarsening algorithm to balance efficiency and performance at arbitrary ratios.
>
> **Approximate coarsening**:  (1) Exact coarsening alone achieves ratios of 0.30/0.20/0.33 on Cora/Citeseer/DBLP. Therefore, Theoretical guarantees hold for any ratio larger than these ratios. (2) We introduce ApproximateCoarsening to support arbitrary coarsening ratios (e.g., 0.1), because every graph has an inherent minimal scale below which topology features cannot be preserved. This phase handles extreme compression demands beyond this limit.
>
> **Tuning time**: \theta_1 is the only tuned parameter, serving to balance efficiency and topology preservation. Other parameters (r: self-adaptive; \theta_2: fixed) require no tuning. Baselines like GEC tune multiple parameters (e.g., dimension d, subgraph size \tilde{n}).  Since STPGC's parameter search space is smaller, tuning overhead does not compromise its efficiency advantage. We will clarify this in the revision.
>
> **DropEdge**: See Q5.
>
> **Experimental details**: We will move important settings to the main part.
>
> **Neighborhood tree aggregation**:  A neighborhood tree comprises a central node u and its k-hop neighbors (nodes with shortest-path distance < k to u). Proving original tree nodes are not lost in the coarsened graph is equivalent to showing their shortest-path distances to u do not increase. By proving that graph strong collapse and edge collapse operations preserve shortest-path properties, we guarantee that nodes in the original neighborhood tree are retained in the coarsened graph (for graph edge collapse within at most (k+1) hops).
>
> **Response to Questions**:
>
>   **Q1**: Fig. 6 appears non-linear because plotting runtime against node-induced subgraph sampling causes |V| and |E| to grow non-linearly. We re-ran the com-youtube experiments ensuring linear growth of |V| and |E|. The new results confirm runtime scales linearly with |V|. We will update Fig. 6.
>
> |\|V\| Ratio|0.10|0.20|0.40|0.60|0.80|1.00|
> |---|---|---|---|---|---|---|
> |Runtime(s)|34.60|68.02|77.31|96.62|111.96|134.13|
>
> **Q2**:
> (1) Topology Advantage: Topology (homotopy equivalence) preserves global structures (e.g., rings, connected components) that other methods can destroy. Preserving these global features stabilizes the local neighborhoods that GNNs rely on. Disrupting topology degrades connectivity: breaking a connected component disconnects 1-hop neighbors entirely, and cutting an edge in a 5-ring stretches a 1-hop distance into 4 hops. Thus, topological features are not merely global invariants; they are essential for maintaining the local node connectivity required for GNN performance.
>  (2) Complexity: STPGC uses efficient, localized graph operations instead of heavy matrix computations. It is the fastest evaluated method (Table 1) and scales to massive datasets (ogbn-products) where spectral baselines suffer OOM issues.
>
> **Q3**:  GCN results for UGC and MPG are below. We will add complete results in revision.
>
> |Dataset|Method|0.50|0.30|0.10|
> |---|---|---|---|---|
> |Cora|UGC|80.1|78.8|76.9|
> |Cora|MPG|80.2|79.2|74.5|
> |Citeseer|UGC|70.0|69.2|68.7|
> |Citeseer|MPG|72.2|71.4|71.0|
> |DBLP|UGC|84.5|83.0|74.3|
> |DBLP|MPG|84.8|80.3|75.0|
>
> **Q4**:
> The major division between the topological/spatial perspective is whether the homotopy equivalence is preserved, just as spectral/spatial methods differ on preserving graph spectrum. Homotopy equivalence theoretically guarantees topological feature preservation (e.g., rings). Conversely, spatial methods (e.g., Affinity GS) merge nodes solely based on proximity, risking ring disruption.
>
> **Q5**: We re-ran experiments without DropEdge. STPGC still outperforms baselines in 46/60 settings. Complete results will be in the revision.
>
> |Dataset|Ratio|GCN|APPNP|SAGE|SAINT|
> |---|---|---|---|---|---|
> |Cora|0.5|81.9|84.2|82.7|82.2|
> |Cora|0.3|81.5|82.7|82.6|82.3|
> |Cora|0.1|82.8|85.1|83.2|81.2|
> |Citeseer|0.5|71.6|71.0|71.4|72.5|
> |Citeseer|0.3|71.7|71.3|72.8|72.2|
> |Citeseer|0.1|71.4|72.9|73.5|74.5|
> |DBLP|0.5|84.4|85.8|82.1|81.8|
> |DBLP|0.3|84.2|84.5|81.7|82.2|
> |DBLP|0.1|82.0|82.9|81.4|83.1|
> |ArXiv|0.5|70.3|62.5|67.7|67.9|
> |ArXiv|0.3|68.4|61.0|65.5|64.4|
> |ArXiv|0.1|64.6|58.7|63.1|64.7|
> |Products|0.5|75.1|65.7|73.9|73.8|
> |Products|0.3|74.2|64.8|72.7|72.5|
> |Products|0.1|70.3|60.5|67.9|67.7|

---

> > ### Author Rebuttal · Reviewer_4T4m · 2026-04-04
> >
> > I appreciate the author's response, in which they addressed numerous points regarding both the experiments and the motivation behind the work. The author supplemented their submission with corresponding graph coarsening experiments, thereby demonstrating the performance of their algorithm. Furthermore, they provided clarifications regarding the motivation, highlighting the significance of topology for GNN algorithms as well as the broader impact of this research. Although I still harbor some reservations regarding the distinction drawn between the topological and spatial levels, I consider the response as a whole to be reasonable. I will raise my score.

---

> > > ### Author Response · Authors · 2026-04-04
> > >
> > > Thank you very much for your positive feedback and for raising your score. We are delighted that the new experiments and clarifications resolved your main concerns.
> > >
> > > We appreciate your note regarding the topological versus spatial distinction. We will include a visual example in the revision to more concretely illustrate the boundary drawn between the topological and spatial levels.

---

### Official Review · Reviewer_mm4f · 2026-03-12

**Soundness:** 3
**Presentation:** 3
**Significance:** 3
**Originality:** 3
**Overall Recommendation:** 4
**Confidence:** 4

**Summary:**

The paper introduces a scalable graph coarsening approach that preserves topological properties in the compressed graph. It builds upon prior work [R1], which uses the concept of elementary collapse and is among the first methods addressing topology-preserving graph coarsening. The proposed method consists of three operations: GStrongCollapse (iterative removal of dominated nodes), GEdgeCollapse (removal of dominated edges), and Neighborhood Coning (targeted edge insertion), which together form STPGC. To achieve arbitrarily small coarsening ratios, the authors propose a two-stage framework: first applying STPGC for topology-preserving coarsening, followed by ApproximateCoarsening, which relaxes the collapse conditions. The authors also show that these operations preserve the shortest-path distances between the remaining nodes. Experimental evaluation is conducted on small-, medium-, and large-scale datasets.

**Compliance With Llm Reviewing Policy:**

Affirmed.

**Final Justification:**

All my concerns were addressed by the authors and the authors have completed the experiments on the heterophilic datasets. Although, with the results I think the method is primarily restricted on homophilic datasets. So I will maintain score.

**Key Questions For Authors:**

Refer weakness

**Limitations:**

Yes, there is an impact statement. However, the authors should discuss the limitations of the framework.

**Strengths And Weaknesses:**

Strengths:
1. The problem is well-motivated and the framework description is well written.
2. The results in Table 1 clearly highlight the efficiency of the framework.
3. The authors have included theoretical results to show that each step does not change the shortest path distance.
4. The framework produces a coarsened graph with consistent topological property preservation over a large range of coarsening ratios identified by the Betti number.

Weakness:
1. The paper shows that the individual operations preserve shortest-path distances between remaining nodes. However, it remains unclear how these guarantees extend to the overall framework and its impact on GNN performance. The paper claims preservation of GNN receptive fields.
2. How well the method performs on coarsening heterophilic graphs?
3. How does the ApproximateCoarsening step perform with different choices of r and how is it tuned?

---

> ### Author Rebuttal · Authors · 2026-03-31
>
> **Response to Weaknesses**:
>
> **W1**:
> (1) **Individual operations and the overall framework**.
> The overall exact coarsening framework is an iterative, intertwined composition of graph strong collapses and edge collapses. We explain how their cumulative effects extend to the overall framework: Graph strong collapse: As  proven in Lemma 3.3, a graph strong collapse strictly guarantees that the shortest-path distances between remaining nodes do not increase. Its cumulative effect remains strictly bounded for preserving shortest path distance, which can be extended to the overall exact coarsening. Graph edge collapse: As proven in Lemma 3.4, a single graph edge collapse can increase the distance by at most 1. However, this penalty does not recursively accumulate along a long shortest path across the graph for two  reasons: First, an edge collapse on (u,v) can only occur if it is dominated by a common neighbor w. This  restricts edge collapses  to locally dense structures (e.g., cliques). Second, we use edge collapse as a catalyst for strong collapse. Nodes that become dominated after a graph edge collapse will be deleted in the next round of graph strong collapse, offsetting the potential distance increase of 1 introduced by graph edge collapse.
> We acknowledge that this claim holds only in the Exact Coarsening phase of the overall framework, and we will clarify this in the revision.
> (2) **Impact on preserving GNN performance**.
> In message-passing GNNs, the receptive field of a node refers to the set of neighborhood nodes from which it can aggregate messages (typically its 2-hop or 3-hop neighborhood). If the shortest-path distance between a neighbor node and a central node is k in the original graph, this neighbor  falls within the central node's k-hop receptive field. If this shortest-path distance does not increase in the coarsened graph, the neighbor is mathematically guaranteed to remain within that $k$-hop neighborhood. Therefore, by ensuring that shortest-path distances between remaining nodes do not increase, the original receptive field of a node is effectively preserved within its receptive field in the coarsened graph.
>
> **W2**: We evaluated three heterophilic graph datasets using an heterophilic GNN (GPRGNN [1]). The results are shown below. Due to time constraints, we have not evaluated on other datasets. We will include complete results and discussions in revision.
>
> |Method|Texas|Wisconsin|Cornell|
> |---|---|---|---|
> |Original|56.5|52.5|44.1|
> |STPGC|57.8|54.6|45.2|
> |GEC|58.1|54.1|46.2|
> |Var.Node|57.6|53.5|44.3|
> |Var.Edge|58.6|52.5|41.9|
> |Alg.GC|57.0|53.3|43.2|
> |Aff.GS|55.1|52.4|45.1|
> |Kron|59.5|52.0|42.7|
>
> [1]. Chien E, Peng J, Li P, et al. Adaptive Universal Generalized PageRank Graph Neural Network. ICLR 2021.
>
>
> **W3**:  We clarify that we did not tune the parameter r. In fact, r is an adaptive parameter that does not require manual tuning. As shown in lines 1 and 18-19 of Algorithm 4, r is 0 during the exact coarsening phase. During the approximate coarsening phase, it automatically increments by 1 only when the number of deleted nodes in a single round is less than \theta_2, ensuring that the coarsening process can  proceed.
>
>
> **Response to Limitations**:
>   Thank you for the suggestion. A limitation is  that we focus  on topological structure to build the graph coarsening algorithm. Because we rely on simple heuristic methods for node features and labels rather than designing dedicated methods, downstream GNN performance might be further improved for specific tasks. We will add discussions on limitations in the revision.

---

> > ### Author Rebuttal · Reviewer_mm4f · 2026-04-03
> >
> > Thank you for the clarifications provided in the rebuttal. The authors have addressed the concern regarding theoretical guarantees, confirming that these do not hold for both algorithms. Based on the additional heterophilic results, the proposed method does not appear to perform as strongly in heterophilic settings.
> > Based on these and other reviews, I will maintain my original score.

---

> > > ### Author Response · Authors · 2026-04-07
> > >
> > > We sincerely thank you for acknowledging our rebuttal and maintaining your positive evaluation. We will explicitly discuss the theoretical guarantees and the method's limitations in heterophilic settings in the revision. Thank you again for your thoughtful guidance.

---

### Official Review · Reviewer_a7fw · 2026-03-13

**Soundness:** 2
**Presentation:** 2
**Significance:** 3
**Originality:** 3
**Overall Recommendation:** 3
**Confidence:** 3

**Summary:**

The paper proposes Scalable Topology-Preserving Graph Coarsening (STPGC), a framework inspired by concepts from algebraic topology. The method introduces three main algorithms: Graph Strong Collapse, Graph Edge Collapse, and Neighborhood Coning, which iteratively remove dominated nodes and edges while preserving topological features such as connectivity and cycles.

The authors mention that their method preserves homotopy equivalence and GNN receptive fields, which theoretically maintains downstream model performance. An approximate variant is also proposed to achieve flexible coarsening ratios. Experiments on node classification tasks demonstrate improved scalability and performance compared to prior topology-preserving methods.

**Compliance With Llm Reviewing Policy:**

Affirmed.

**Key Questions For Authors:**

1. How does STPGC perform on graphs with millions of nodes, where neighborhood checks and dominance detection may become expensive?

2. Sensitivity to parameters: The algorithms rely on thresholds such as (\Theta 1, \Theta 2) and the relaxation parameter r. How sensitive is the method to these choices in practice?

3. Have the authors evaluated the method on other graph learning tasks such as graph classification or link prediction?

4. When approximate coarsening is used, how often are important topological features lost, and how does this affect downstream GNN performance?

5. How does STPGC compare with widely used spectral or diffusion-based coarsening methods in terms of both runtime and accuracy?

**Limitations:**

The paper does not explicitly discuss the limitations or potential societal impacts of the work.

Although the method is primarily algorithmic, it would still benefit from a clearer discussion of its limitations.

For example, the algorithms require neighborhood inclusion checks and dominance detection, which may become computationally expensive on dense graphs or graphs with very high degree nodes.

Also, the empirical evaluation is limited to node classification tasks, and it remains unclear how well the method generalizes to other graph learning settings.

**Strengths And Weaknesses:**

1. Soundness:

The proposed method is grounded in ideas from algebraic topology, particularly strong collapse and edge collapse, which are extended to the graph setting. The theoretical arguments regarding homotopy equivalence and receptive field preservation are reasonable and supported by formal lemmas.

However, several aspects could be strengthened. First, the theoretical guarantees mainly focus on structural preservation properties rather than direct guarantees about learning performance.

Also, while experiments show improvements over GEC and other baselines, the empirical evaluation is mostly limited to node classification benchmarks, leaving open questions about performance on other graph tasks such as link prediction or graph classification.

2. Presentation:

The paper is generally well organized and the motivation is clearly stated. The transition from algebraic topology concepts to graph algorithms is logical, and the pseudocode for the main algorithms is helpful.

That said, the presentation could still be improved. In particular, the figures could have detailed captions and larger label font sizes, which would make them easier to understand.

Some of the theoretical explanations are also fairly dense. Adding more intuitive explanations or small examples could make the paper easier to follow for readers who are not familiar with algebraic topology.

3. Significance:

Graph coarsening is an important problem in large-scale graph learning. Preserving topological structures such as cycles and connectivity is a meaningful direction. If the proposed method proves robust across different graph learning tasks and datasets, it could be useful for efficient training on large graphs while maintaining predictive performance.

However, the impact may depend on whether the method generalizes well beyond the specific benchmarks evaluated in the paper.

4. Originality:

While the work builds on existing ideas from topology and graph coarsening, the combination of these concepts into a scalable algorithmic framework appears to be a meaningful and original contribution.

---

> ### Author Rebuttal · Authors · 2026-03-31
>
> **Response to Weaknesses**:
>
> **Structural preservation and performance**: Structural preservation is an important factor for GNNs' learning performance. The core learning mechanism of GNNs relies on message passing over multi-hop local neighborhoods (i.e., the receptive field). As proven in Lemma 3.3 and Lemma 3.4, our algorithms guarantee that the shortest path distances between nodes do not increase (or increase by at most 1 in the case of edge collapse). This ensures that the receptive field of each node remains largely intact after coarsening. By preserving the underlying computational graph that the GNN relies on for feature aggregation, STPGC inherently protects the model's input information on coarsened graphs. We acknowledge that the actual performance of GNNs relies on multiple factors, including node features in addition to the graph structure. However, we focus specifically on maintaining GNN performance during graph coarsening from the perspective of preserving the topological structure.
>
> **Link prediction & graph classification**: We have conducted additional experiments on link prediction and graph classification (MUTAG/Peptide).  Due to space limitations, please refer to the response to reviewer w3GC W3&Q1 for results. We will add complete results to the revision.
>
> **Presentation**: We will enlarge figure labels, detail captions, and add intuitive examples (e.g., simplicial complexes, homotopy equivalence) in the revision.
>
> **Additional benchmarks**: We have conducted additional experiments on link prediction tasks, the MUTAG/Peptide graph classification dataset (response to reviewer w3GC W3), heterophilic graphs (response to reviewer mm4f W2), and long-range benchmark Peptide functions (response to reviewer w3GC Q1).
>
> **Response to Questions**:
>
>   **Q1**: We have evaluated STPGC on several million-scale graphs, including ogbn-products, LiveJournal, Youtube, cit-Patent, and Flixter. Our Scalability Analysis (Sec. 4, p. 7) demonstrates that runtime and memory usage scale smoothly, with STPGC consistently achieving substantial speedups over GEC (up to 37x on cit-Patent).
>
>   **Q2**: For $\theta_1$, we included a parameter sensitivity study (p. 7, Figure 5), which evaluates its impact on both accuracy and runtime across different coarsening ratios. We clarify that the parameter in Figure 5 was erroneously labeled as $\theta$ instead of $\theta_1$. Our results shows that after $\theta_1$ exceeds a certain threshold (e.g., 10 for Cora), the accuracy becomes relatively stable. Parameter $r$ is self-adaptive, requiring no tuning. $\theta_2$ is a fixed parameter that does not directly affect performance. It only controls how fast $r$ increases, which balances topology preservation and efficiency. Since $\theta_1$ already handles this balance, $\theta_2$ needs no manual tuning.
>
>   **Q3**: Please refer to Response to Weaknesses.
>
>   **Q4**: (1) Frequency of Topological Feature Loss. The threshold for entering ApproximateCoarsening depends on the dataset's inherent topology (e.g., a ratio of 0.3 for Cora and 0.2 for Citeseer; Figure 4). Even in this relaxed phase, STPGC preserves topological features significantly better than baselines. (2) GNN Performance. All datasets use exact coarsening at a 0.5 ratio and approximate coarsening at 0.1. Despite entering the approximate phase at 0.1, accuracy on Cora, Citeseer, and DBLP remains comparable to 0.5, showing the phase effectively maintains GNN performance. The performance drop on ogbn-arxiv and ogbn-products at 0.1 is possibly due to supernode feature mixing under such a low coarsening ratio.
>
>   **Q5**: We evaluated widely used spectral baselines (Variation Neighborhoods, Variation Edges, kron, FGC). STPGC achieves superior or competitive accuracy (Table 2, p. 7) and is consistently faster than all spectral baselines (Table 1, p. 6).
>
> **Response to Limitations**:
>
>   **Limitations and impacts**:  We will add  discussions  in the revision. Limitation: STPGC focuses solely on topology, using heuristic feature/label aggregations. Task-specific feature processing could further improve downstream performance.
>     Societal Impact: By accelerating GNN training on massive datasets, STPGC reduces energy consumption, contributing to "Green AI."
>
>   **Computation cost**: While dense graphs and high-degree nodes pose computational challenges, we implemented specific optimizations to efficiently address them: early pruning of candidate checks (p. 3, Alg. 1, line 8), hash-based membership tests (p. 3, discussion below Alg. 1), lazy deletion (p. 3, discussion below Alg. 1), and degree thresholding (p. 3, Alg. 1, line 6). Consequently, the time complexity for high-degree nodes is  bounded by the degree threshold $\theta_1$. Our scalability experiments (Fig. 6-8) also demonstrate the efficiency of STPGC on million-scale graphs.

---

> > ### Author Rebuttal · Reviewer_a7fw · 2026-04-04
> >
> > Thank you for your responses.
> >
> > Overall, the rebuttal enhances the clarity of the work and addresses the majority of my concerns. However, a few areas would still benefit from further clarification. I will maintain my original score.

---

> > > ### Author Response · Authors · 2026-04-04
> > >
> > > We sincerely thank you for reviewing our rebuttal and for acknowledging that the clarity of our work has been enhanced and the majority of your concerns are now fully resolved.
> > >
> > > We noticed your comment that a few areas would still benefit from further clarification. Could you please specify which areas you are referring to? If possible, please do not hesitate to let us know, we would be more than happy to provide further clarification or additional responses.

---

### Decision · Program_Chairs · 2026-04-30

**Decision:**

Accept (regular)

**Comment:**

In this paper, the authors introduce a graph coarsening method with an objective of receptive-field preservation, which they claim is useful for preserving GNN performance. The reviewers globally appreciated the underlying idea and were generally satisfied by the detailed rebuttals and answers provided by the authors. They however raised some remarks asking for clarification about the underlying motivation for the method (i.e. the link between receptive field and GNN performance), in particular compared to other methods that exhibit performance guarantees (e.g. references by reviewer 4T4m). Still the overall view of the paper remains positive, and I encourage the authors to carefully include their response to reviewers in the revised version of the paper.